# No evidence that hominin dispersal across Eurasia was part of a wider turnover in mammal distributions

Jijia Sun [1,2] ✉, Ignacio de la Torre[3] & Faysal Bibi [1] ✉

The drivers and consequences of hominin dispersals out of Africa remain debated. The spatial and temporal distribution of large mammal faunas contemporaneous with early *Homo* provides direct evidence for their ecological context and impact. In this study, we conduct taxonomic and functional similarity analyses on fossil and extant Eurasian and African large mammal communities of the last 10 Ma. We test two hypotheses: 1) the dispersal of hominins across Eurasia around or shortly after ~2 Ma was part of a wave of faunal dispersals out of Africa; 2) the arrival of hominins at Eurasian sites coincided with major changes in the functional structure of large mammal communities. Our results indicate that hominin dispersals from Africa to Eurasia during the Plio-Pleistocene were not part of a larger faunal expansion. Instead, the most significant faunal interchange during the Plio-Pleistocene occurred between Europe and Asia, while African faunas have mostly remained distinct from Eurasian faunas since ~7 Ma. Our results suggest relative homogeneity in community functional structure across Eurasia and Africa since at least 10 Ma. In contrast to fossil communities, modern Eurasian and African terrestrial large mammal faunas show strong geographic functional structure, which might reflect the selectivity of Late Pleistocene extinctions.

For most of their evolutionary history, hominins were restricted to Africa. The driving factors behind their dispersal out of Africa, as well as their influence on contemporaneous Eurasian mammal communities, are subjects of interest and debate. There were at least two major out-of-Africa events. The first was during the Early Pleistocene, possibly ~2 Ma based on artifacts from Shangchen, China[1] and Grăunceanu in Romania dated to 1.95 Ma[2]. From around 1.8 Ma, there is conclusive evidence of the presence of hominins in Eurasia[3,4] based on fossils from Dmanisi, Georgia[5]. Shortly thereafter, stone artifacts or hominin remains are reported from the Yuanmou Formation in southern China, dated to 1.7 Mya[6], Majuangou III and Shangshazui in the Nihewan Basin, dated to 1.6–1.7 Mya[7,8], Ubeidiya in the Levant dated to 1.4 Ma[9] as well as Atapuerca and Orce in Spain, dated to 1.1–1.3 Mya[10].

Different scenarios have been proposed to explain Early Pleistocene dispersals, including intrinsic evolutionary factors[11,12] (e.g., brain expansion and technological advances) and external factors[13–20] (e.g., environmental conditions including climate change as well as tectonics, and accompanying faunal dispersal). There is significant speculation about the ecological context and adaptive capacities of these early hominin dispersers. Messager et al. (2011), for instance, have claimed that, during the Early Pleistocene, hominins gradually increased their capacity of adaptation to diversified environments[4].

A second major out-of-Africa event occurred during the Late Pleistocene when *Homo sapiens* dispersed outside Africa. There are different hypotheses on possible driving factors of this dispersal, including environmental, fauna, or cognitive changes and technological advances[21–25].

[1]Museum für Naturkunde, Leibniz Institute for Evolution and Biodiversity Science, Berlin, Germany. [2]Faculty of Life Sciences, Humboldt-Universität zu Berlin, Berlin, Germany. [3]Pleistocene Archaeology Lab, Instituto de Historia, CSIC-National Research Council, Madrid, Spain. ✉e-mail: jijiasun7@gmail.com; faysal.bibi@mfn.berlin

The hypothesis that Pleistocene hominins might have followed other large mammals out of Africa is based on assumptions of predator-prey relationships of hominins and herbivores, or a coevolutionary relationship between hominins and large carnivores[15,26,27]. The spatial and temporal distribution of large mammal faunas contemporaneous with early *Homo* could help us understand the context and impact of hominin dispersal out of Africa. Large mammals that are believed to have dispersed into Eurasia during the Plio-Pleistocene include *Theropithecus, Panthera, Megantereon, Crocuta, Pachycrocuta, Hippopotamus, Palaeoloxodon, Hippotragus, Oryx, Damalops, Pelorovis, Kolpochoerus, Parahyaena, Giraffa, Vishnukobus* and *Potamochoerus*[19,28]. During their first dispersal, *H. erectus* might follow large herbivores[15], with cervids, bovids, and likely suids serving as important prey in Southeast Asia[29].

Among large mammalian species, carnivores might have played a significant role in hominin dispersals[12]. Hominins might have incurred in an intense competition with carnivores for both resources and space[30]. Lewis and Werdelin (2010) proposed that large carnivores likely influenced hominin dispersal based on a significant overlapping of diet and habitat[31]. Rodríguez et al. (2023) suggested that early hominins were capable of competing with giant hyaenas for carcasses generated by saber-toothed felids[32]. Thus, whether hominins played a role as predators or scavengers was possibly determined by the competition stress from carnivores[30]. Among carnivores, *Panthera gombaszoegensis, Pachycrocuta brevirostris*[33,34], *Crocuta crocuta*[35], *Megantereon whitei*[34] and *Panthera leo*[36] dispersed into Eurasia during the Early Pleistocene[33-36]. But whether *Pachycrocuta* and *Crocuta* originated in Africa is still unclear[36], and it should be taken into account that carnivores tend to have a wider geographic distribution than herbivores and can disperse quickly and widely[37,38].

Nonetheless, some hold the view that faunal exchanges were not correlated with hominin dispersals out of Africa[27,39]. Analysis based on the first appearance of African and Eurasian mammals showed little evidence for waves of dispersal during Plio-Pleistocene[19]. Tong (2011) pointed out that, although there was some faunal exchange between Africa and Eurasia during the Neogene, there are few taxa in northern China that directly originated from sub-Saharan African faunas during the Pleistocene[36]. And some genera might also have dispersed into Africa from Eurasia, such as *Equus, Nyctereutes, Lycaon*, and *Antilope*[19].

Additionally, there are questions about the impact of hominins on Eurasian faunal communities after their arrival from Africa. Some researchers suggested that the reduction of functional richness of the large carnivore (>21.5 kg) guild starting from 2 Mya may have been caused by competition with hominins[40]. In contrast, other studies found evidence for long-term declines in African large mammal diversity and abundance since ~4 Ma, precluding a role for hominins[41,42]. Hominin dispersals in the Early and Middle Pleistocene were found not to be correlated with faunal turnover[39]. During the Late Pleistocene dispersal of *H. sapiens*, hunting activities are likely to have had a significant influence on the extinction of large mammals[43-46]. Studies suggested that Late Quaternary large mammal extinction with selectivity for large body sizes appeared on all continents except Africa and the unusual body-size selectivity is correlated with human arrival, through both direct (hunting) and indirect (competition and habitat alteration) effects[47]. Mammal extinctions in Africa during the Late Pleistocene and Holocene were suggested to be driven by environment changes[48]. Megafaunal declines in Europe have been proposed to be driven by *H. sapiens*[49]. A notable decline of species richness after 10 Ma was also found in European fauna[50].

Based on the information above, we analyze Eurasian and African large mammal faunas of the last 10 Ma to obtain a continental-scale perspective of faunal change and dispersal in this study. We examine the taxonomic and functional similarity of fossil communities to better understand the faunal context and potential impact of hominin dispersals out of Africa. We test two hypotheses: (1) the dispersal of hominins across Eurasia around or shortly after ~2 Ma was part of a larger wave of faunal dispersal out of Africa; (2) the arrival of hominins in Eurasia resulted in major changes in the functional structure of large mammal communities. Our results show no evidence of faunal dispersal waves out of Africa and a rather similar functional community structure across Eurasian and African fossil sites, in contrast to extant communities, which show strong latitudinal differences.

## Results

### Fossil taxonomic analysis

The result of silhouette coefficient analysis indicated the number of k should be 3 for the fossil taxonomic clustering (Supplementary Fig. 1A), while the average membership degree (AMD) analysis indicated just 2 groups in the data (Supplementary Fig. 1E). With k = 3, the three resulting clusters basically comprised an African cluster, a mainly Late Miocene cluster and a mainly Pliocene to Pleistocene Eurasian cluster (Fig. 1A). From 10 to 7 Ma, all communities across Eurasia and Africa, including two African communities (Chorora site and Ngeringerowa site), belonged to a single cluster (cluster 1 in Fig. 1). The ten most frequently occurring genera were *Hipparion, Gazella, Hippopotamodon, Palaeotragus, Tragoportax, Adcrocuta, Chilotherium, Cremohipparion, Hyaenictitherium* and *Ictitherium*. The Plio-Pleistocene Eurasian cluster (cluster 2 in Fig. 1), characterized by *Cervus, Equus, Sus, Ursus, Panthera, Canis, Stephanorhinus, Mammuthus, Bison* and *Rhinoceros*, appeared between 7 and 5 Ma in western Eurasia, then dispersed to replace the Miocene cluster entirely across Eurasia by 3 Mya. Late-surviving genera of the Late Miocene cluster in Asia included *Canis, Gazella*, and *Hipparion*. Around the same time, a distinct African cluster began to develop at 7-6 Ma (cluster 3 in Fig. 1). The ten most frequently occurring genera in this cluster were *Tragelaphus, Gazella, Aepyceros, Kobus, Elephas, Kolpochoerus, Ceratotherium, Hippopotamus, Giraffa* and *Equus*. Overall, from this time on, African faunas remained basically different from those of Eurasia and were not affected by the faunal turnover that took place in Eurasia between 6 and 3 Ma. We also visualized the clustering by epoch, with similar results (Supplementary Fig. 2A).

The chronofauna similarity analysis of the Shanshenmiaozui (SSMZ) fauna revealed a peak similarity of both European and Asian communities during the Early Pleistocene (Fig. 2A). In contrast, persistently low similarity of the African fauna with SSMZ confirms little to no mixing of genera between Africa and eastern China at this time. The chronofauna analyses of other communities from the Pleistocene of Spain (Atapuerca TDW4 & TDE5), Miocene of Anatolia (Akkasdagi), and Pleistocene of Kenya (Karari Ridge 2) confirm the high similarity of Asian and Eurasian faunas during the Late Miocene (Akkasdagi) and their later divergence during the Pleistocene (Atapuerca), and the highly endemic nature of African Plio-Pleistocene faunas (Karari Ridge) (Fig. 2B–D).

The nMDS analyses of all communities by genera confirmed the high overlap of Eurasian fossil communities, and the distinctiveness of African communities since the Miocene (plotted by epoch in Fig. 3A). A stress value of 0.17 indicates an acceptable ordination fit, with the major patterns being represented. Similarly, the correspondence analysis of communities by genera confirmed the strong continental taxonomic differentiation between Eurasia and Africa (Fig. 4A). In both the nMDS and correspondence analyses, the first axis separates between older (to the right) and younger (left) Eurasian communities, confirming the turnover between taxonomic clusters 1 and 2 shown in Fig. 1A. In both analyses the second axis consistently distinguished Eurasian from African communities.

### Fossil functional analysis

The same method was applied to the functional clustering. The silhouette coefficient analysis indicated that the best number of k should be 2, while the AMD analysis indicated 3 clusters (Supplementary Fig. 1B, F). For a better comparison between the taxonomic and

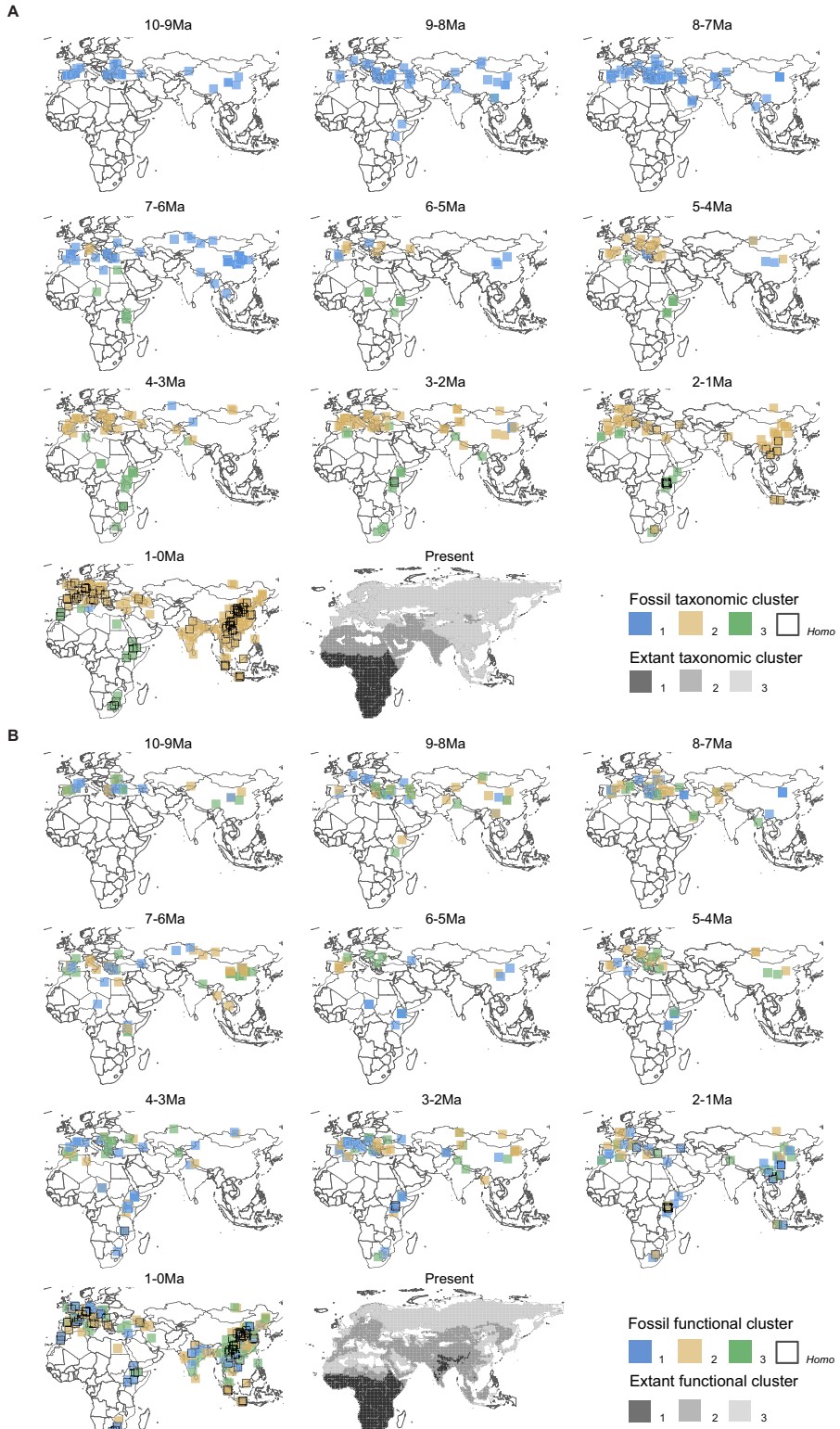

**Fig. 1 | Clustering of African and Eurasian fossil and extant large mammals.**
**A** Taxonomic clustering shows strong temporal and geographic differences across continents in fossil data (shown in 1 myr intervals). Miocene faunas (cluster 1) survive longest in Asia, to be replaced by genera that first emerged between 7 and 4 Ma in Europe (cluster 2). African faunas (cluster 3) remained distinct from 7 mya on. Extant communities similarly show strong latitudinal structure. **B** Clustering of fossil communities by functional traits (diet, locomotion, size) shows no apparent geographic structure, in contrast to extant communities, which have strong latitudinal structure. All analyses were conducted at the genus level, with the number of clusters set to three.

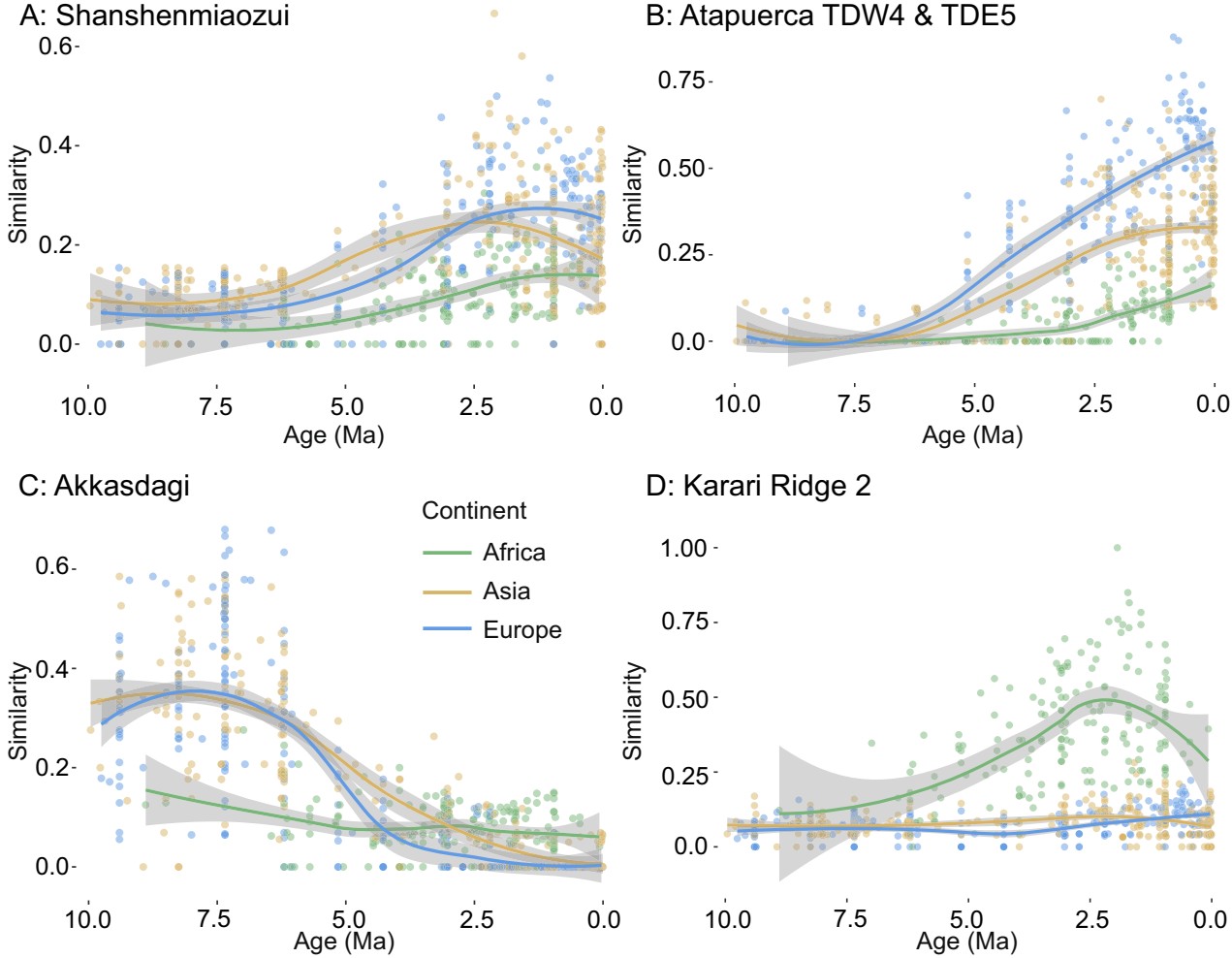

**Fig. 2 | Taxonomic similarity chronofauna plots.** Taxonomic similarity chronofauna plots. Shown is genus-level taxonomic similarity to (**A**), Shanshenmiaozu (SSMZ, China), **B** Atapuerca TDW4 & TDE5 (Spain), **C** Akkasdagi (Turkey), **D** Karari Ridge 2 (Kenya), and other fossil sites in Eurasia and Africa. All analyses indicate African faunas remained distinct from those of Eurasia. The gray shaded area is the 95% confidence interval.

functional analyses, k was set to be 3. The resulting three functional clusters show high overlap with no apparent structure across temporal intervals or geographic regions. Visualizing the data by epoch shows a similar result (Supplementary Fig. 2B). A sensitivity analysis using just two traits (body mass, diet) showed a similar result, indicating that any uncertainty of locomotion assignment was not significantly biasing the results (Supplementary Fig. 3).

The functional nMDS analysis based on total dissimilarity similarly showed no patterns of difference among the three continents (Fig. 3B). Stress value was 0.087.

The correspondence analysis of community by traits also showed a high overlap of sites across all three continents. Ground dwelling herbivores between 45 kg and 10 tons were the most common large mammals on all three continents. Trait distributions therefore do not show major differences among continents in the fossil data.

### Extant taxonomic analysis
Both the silhouette coefficient and AMD analyses of the extant taxonomic data showed a best cluster number of 3 (Supplementary Fig. 1C, G). Taxonomic clustering with k = 3 showed pronounced latitudinal variation separating the Old World into northern and southeastern Eurasia, North Africa to South Asia, and sub-Saharan Africa (Fig. 1A). Analyses using variations of the PHYLACINE dataset (including estimates of where species would live without anthropogenic pressures)

showed a similar pattern with only slight differences to the IUCN data shown in Fig. 1A (Supplementary Fig. 4).

### Extant functional analysis
The silhouette coefficient analysis showed a result of 2 for the functional clustering, while in the AMD analysis the result showed a result of 3. (Supplementary Fig. 1D, H) The functional clustering also showed a latitudinal pattern, with clusters uniting northern Eurasia with North Africa, central Asia with the Middle East as well as part of southern Europe, and South and Southeast Asia with sub-Saharan Africa (Fig. 1B: present). Similar to the fossil functional analysis, we conducted an analysis based on 2 traits (body mass, diet) for the extant IUCN data in the study. The result showed a similar pattern with minor differences in the distributions. We also conducted the analysis based on both PHYLACINE current ranges and present natural ranges for the functional clustering. The result showed differences from clustering based on IUCN data. One cluster basically occupied north Africa while the other two clusters occupied Eurasia in different regions. The pattern, however, still showed a latitudinal distribution of different clusters here (Supplementary Fig. 4).

## Discussion
### No evidence for waves of faunal dispersal out of Africa
The first hypothesis we sought to test is whether hominins arriving in northern China ~2 Ma were part of a larger wave of faunal dispersal

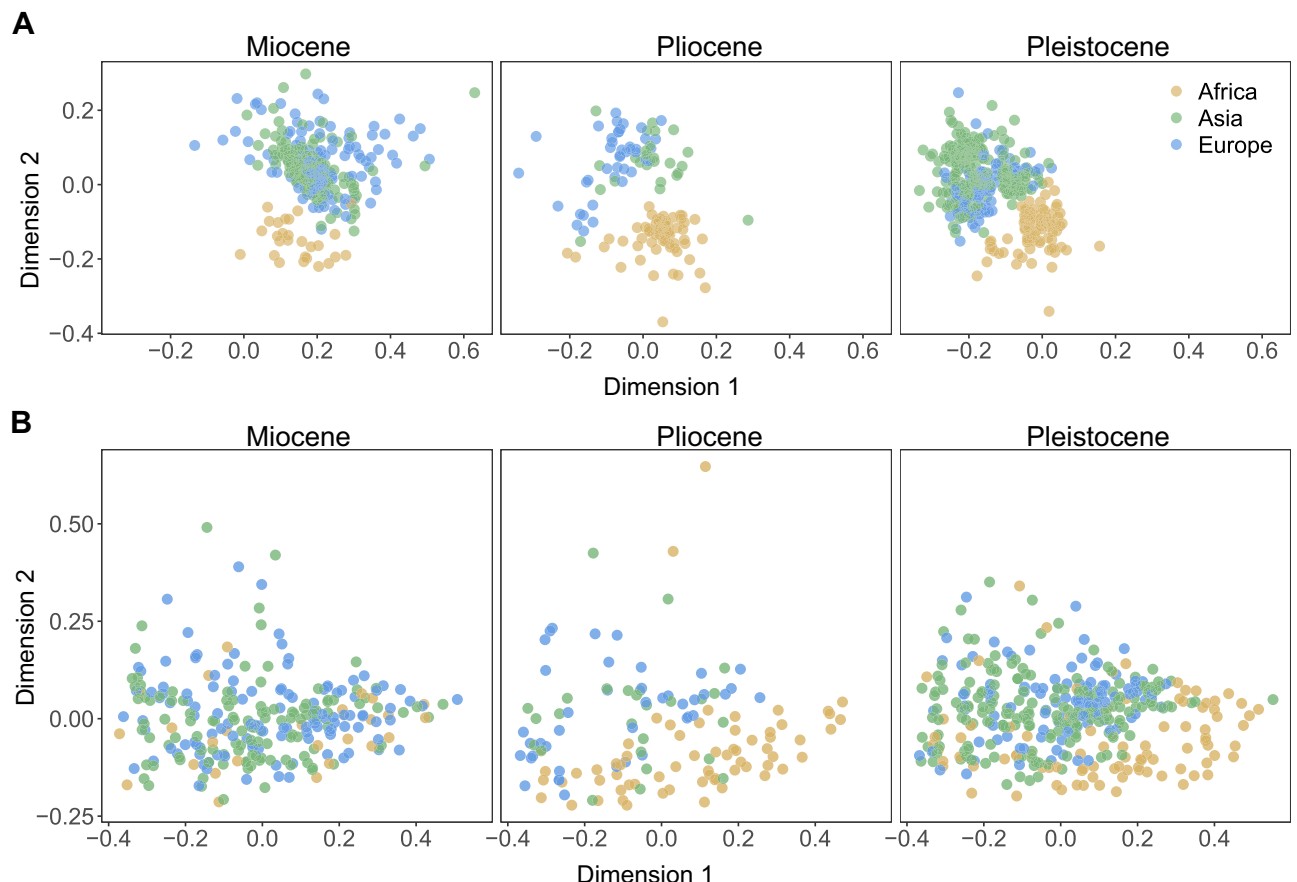

**Fig. 3 | Non-metric multidimensional scaling analysis.** Each point is a paleo-community, colored by continent. **A** Genus-level taxonomic dissimilarity, confirming a consistent separation of African from Eurasian faunas through time.

**B** Functional dissimilarity, showing a lack of continental differentiation. Analyses were conducted using all data, then visualized by epoch.

from Africa to Eurasia. Our analysis of taxonomic similarity at the genus level does not support the presence of a major wave of large mammal dispersal out of Africa at this time, or at any other time during the last 10 Ma. Though African data for the Late Miocene is sparse, communities at 9-8 Ma show greater similarity to contemporaneous Eurasian communities than later African ones, perhaps supporting the proposition of an Old World Savanna Paleobiome[51]. However, by 7-6 Ma African faunas cluster more closely with later African communities than Eurasian faunas. Around the same time in Eurasia, a major faunal interchange took place whereby genera appearing first in Europe at 7-5 Ma totally replaced more archaic Asian taxa by the late Pliocene. African and Eurasian faunas remained highly dissimilar throughout the Pliocene and Pleistocene. The results from the nMDS and correspondence analyses were consistent with the taxonomic clustering analysis, showing a high similarity of European and Asian communities, and a consistent separation of African communities (Figs. 3A, 4A).

Our results support previous studies that found a lack of correlation between hominin and faunal dispersals during the Early Pleistocene[39]. Claims that *H. erectus* passively followed[15] the dispersal routes of large herbivores during their dispersal out of Africa are not supported by our continental-scale perspective. If anything, African and Eurasian faunas were most dissimilar during this time (Figs. 1–4). The dispersal of *Homo* out of Africa therefore likely not a result of major extrinsic (e.g., climatic) factors that would have significantly affected other large mammals as well. Any driving factors for *Homo* and other dispersers during this time were likely lineage-specific, such as new technological, behavioral, or dietary adaptations. We therefore

conclude that the dispersal of *Homo* out of Africa, if driven by any factors at all, occurred in a context largely specific to *Homo*.

### No evidence for changes in functional community structure in the fossil record

The second hypothesis we sought to test was whether the arrival of hominins in Eurasia coincided with major changes in the functional structure of communities, as defined by diversity of body size, diet, and locomotion. We found that, over the last 10 Ma, functional communities across Africa and Eurasia showed no clear geographic structure, neither across time nor by continent. We additionally found no evidence for significant changes to functional community structure in Eurasia associated with the arrival of *Homo* from Africa after 2 Ma.

### Evidence for highly altered extant functional community structure

Extant large mammal communities show taxonomic clustering that is latitudinally and continentally structured (Fig. 1A). While this is more nuanced than the fossil geographic clusters, the main distinction of African and Eurasian realms is a strong common feature. This is similar to classical zoogeographic regions based on the distributions of amphibians, birds, and mammals, with identifiable Afrotropical, Saharo-Arabian, and a combination of Palearctic, Sino-Japanese and Oriental regions[52,53](Fig. 1A).

In contrast, the geographic structure of functional similarity of extant communities is starkly different from that of the fossil assemblages. Unlike fossil communities, which showed no apparent geographic functional structure over the last 10 million years, extant

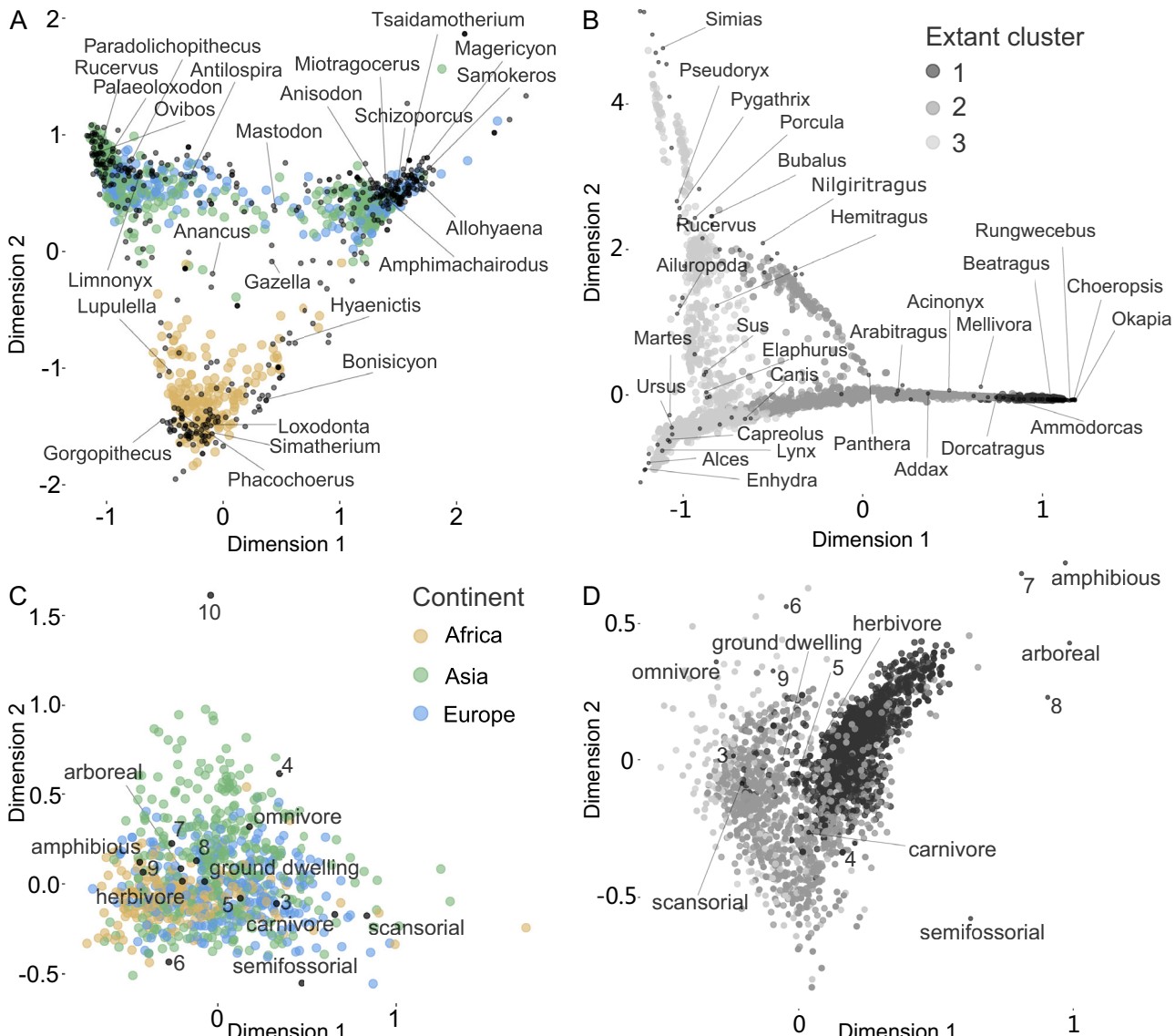

**Fig. 4 | Correspondence analyses of fossil data. A** Analysis of fossil communities by genera. **B** Analysis of extant communities by genera. Only a subset of genera is labeled for readability in (**A** and **B**); **C** Analysis of fossil communities by traits. **D** Analysis of extant communities by traits.

communities reveal clear latitudinal structure, with major differences between mainly Afrotropical, Saharan, and northern Eurasian, and southern Eurasian regions. There are possibly two explanations for these differences: (1) The fossil datasets may be inaccurate or incomplete; or (2) A fundamental rearrangement of functional groups appeared in the time between the fossil and modern dataset, i.e. during the Late Pleistocene and Holocene. We think it is unlikely that this pattern is the result of inaccurate data, particularly given that the attribution of such fundamental traits as size, diet, and locomotion to both fossil and extant genera was in most cases done with a relatively high degree of confidence, allowing also for uncertainty via randomization. The second possibility is more plausible, as much evidence indicates major losses of taxonomic diversity from the Eurasian record during the Middle and Late Pleistocene (e.g., lions, hyenas, proboscideans, rhinos, monkeys, hippos, etc.)[48,54–57]. These taxonomic losses appear to have had disproportionately large functional consequences at higher latitudes, producing the latitudinal variation in functional structure we observed for extant communities. Our correspondence analysis of the extant functional communities shows that the Eurasian regions differ from sub-Saharan Africa mainly in the lack of amphibious and arboreal genera as well as mammals with body mass of 405 to 3675 kg.

This finding suggests that the functional structure appearing in modern large mammal faunas across Africa and Eurasia today may be an artifact of (possibly human-induced) Late Pleistocene and Holocene extinctions. This should be confirmed by further studies, but it adds further supports for the idea fossil communities were functionally non-analogous to those today (even in Africa)[54]. It also provides significant support for the importance of the fossil record to determining natural baselines and forecasting climate change effects[58].

## Methods

We compiled a large dataset of occurrences of fossil large mammals (Artiodactyla, Carnivora, Creodonta, Perissodactyla, Primates, and Proboscidea) from BICAEHGIS, a geographic information system database that includes data from the Paleobiology Database (https://paleobiodb.org/), NOW Database[59], and additional data on 92 Chinese sites and one Romanian site that was entered for this study. All the additional data were collected from published literature. Site age was calculated as the midpoint between the maximum age and minimum

age. If the age was provided as a geological interval (e.g., Late Pleistocene), site age was calculated as the midpoint age of the geological interval. Data is available in Supplementary Data 1. Geographic distribution data for extant large mammals was downloaded from the IUCN Red List website[60] and Phylacine[61,62]. The Phylacine dataset includes species ranges and diversity estimated while accounting for large human impact[61], which documents current or historical ranges, providing an alternative to the IUCN Red List. Both extant and fossil taxonomic data were analyzed at the genus level. Both taxonomic and functional clustering for the extant data used 1° by 1° grid cells as one community. Grid cells are referred to as community throughout this paper. Communities with age uncertainty covering more than one geological epoch (e.g., Pliocene-Pleistocene) were removed. The final dataset includes 570 fossil genera comprising 1513 fossil communities and 204 extant genera comprising 13837 extant communities.

In order to examine community structure changes, we compiled data for three traits: body mass, diet, and locomotion. These traits capture the functional role of taxa in a community, reflecting survival strategies, habitat and resource use, energetic requirements, and trophic level[63,64]. Body mass categories are logarithmic, following Bibi and Cantalapiedra[42], and classified into 10 groups: 1: <1.67 kg; 2: 1.67–5 kg; 3: 5–15 kg; 4: 15–45 kg; 5: 45–135 kg; 6: 135–405 kg; 7: 405–1215 kg; 8: 1215–3645 kg; 9: 3645–10935 kg; 10: >10935 kg. Dietary categories were: herbivore, carnivore, and omnivore. Locomotion categories were: ground dwelling, scansorial, arboreal, amphibious, and semifossorial. Trait data was from the Paleobiology Database, PanTHERIA[65], CarniFOSS[66], Kissling et al.[67], and Faith et al.[54], and from literature describing individual taxa. If a genus has species in different diet categories, all the categories will be assigned to the genus. Data is available in Supplementary Data 2. Taxa belonging to marine habitats were excluded. Synonyms used in this study were provided in Supplementary Data 3. Since the study was based on genus level, one genus might contain more than one body mass or diet category. We used a code to randomize the category used in the study then ran the analyses several times to make sure the randomization choice of the body mass and diet had no significant influence on the final result. Due to size bias against the recovery of very small fossils, taxa belonging to body mass categories 1 and 2 (i.e. <5 kg) were excluded from all analysis. While omnivory can be difficult to define consistently, we note that only 46 out of 625 genera (7.4%) were classified as omnivores, while 32 out of 625 have omnivore in their diet category. Reclassifying genera into or out of the omnivore category would likely have little impact on our results.

The data was analyzed and visualized using R (v 4.3.0)[68] using custom script[69]. We removed all occurrences of indeterminate genera (which are classified as "Indet." or "Gen.") and then kept only communities with at least five genera (e.g.,[51]) to create a presence-absence matrix. Sørensen pair-wise dissimilarity was calculated for beta diversity using the *betapart* package[70,71]. Sørensen pair-wise dissimilarity measures the dissimilarity between two different communities based on their faunal composition. A value of 0 means the two communities are identical while a value of 1 means the two communities shares no common genera. The result consisted of three dissimilarity matrices: total dissimilarity, turnover, and nestedness. Total dissimilarity contains the latter two components without any weight: turnover, which refers to the replacement of taxa between communities; and nestedness, which refers to the degree that a less diverse community is a subset of a more diverse community[70]. Total diversity was used to represent the overall difference among the communities in this study. Communities were clustered using Partitioning Around Medoid (PAM)[72] with the number of clusters (k) set to three, based on the result of silhouette coefficient analysis and average membership degree (AMD) analysis. (Supplementary Fig. 1). PAM is a method similar to k-means but can be applied to a distance matrix. We used the silhouette coefficient as implemented in the *cluster* package[73] to

independently check the best value of k. Additionally, AMD analysis was also applied to the pairwise dissimilarity clustering analysis. AMD analysis is a method for detecting the actual functional groups within the data[74]. The silhouette coefficient and AMD analyses shown in panels A, C, F, G, and H in Supplementary Fig. 1 identified 3 as the optimal number of clusters. Thus, 5 of the 8 analyses supported k = 3 as the best choice. Based on this result, we used k = 3 for all analyses throughout the study. All the data were analyzed together then visualized in 10 intervals of 1 million years duration. Communities with an age uncertainty of more than 2 million years were removed from the visualization but were not removed from the analysis (34 communities). The data was also visualized in three larger time bins: Late Miocene, Pliocene, and Pleistocene. The same method was applied to the extant mammal data compiled from PHYLACINE and IUCN Red List datasets for taxonomic analysis. The result of the IUCN extant data was plotted together with the result of the fossil data for comparison. In the analysis of the PHYLACINE data, we used both current ranges which show current species ranges, and present natural ranges which represent estimates of where species would live without anthropogenic pressures[61] providing an alternative to the IUCN Red List in supplementary. The map used in this study was generated using *rnaturalearth* package in R.

Temporal and geographic dissimilarity change among communities was also investigated using the chronofauna method, which examines how similarity to a single reference community changes across geography and time[51]. The fossil site of Shanshenmiaozui (SSMZ) in the Nihewan Basin, northeastern China, was chosen for this comparison. SSMZ is an Early Pleistocene site (1.7 Ma) with 25 genera (22 of which met our criteria and were included in this study) representing the typical Nihewan Fauna[75]. Although no hominin fossils or stone tools are known from SSMZ, its taxonomic richness and distance from Africa makes it a good reference for cross-continental comparisons. Additionally, chronofauna analysis was also applied to the three medoid communities determined by the PAM analysis, (Akkasdagi, Turkey; Karari Ridge 2, Kenya; and Atapuerca TDW4 & TDE5, Spain). The medoid is the community that shares the lowest total dissimilarity with other communities in the cluster they belong to.

For the functional analysis, the distribution of all three traits for all communities was calculated based on the genus-trait matrix generating a community-trait abundance matrix. This was analyzed using the beta.pair.abund function in *betapart* package[70], employing the Bray-Curtis Index to generate an abundance-based pair-wise dissimilarity matrix. The dissimilarity matrix then was analyzed using PAM with the number of clusters (k) set to three. Silhouette coefficient and AMD analysis were also conducted to examine the best value of k. Analyses using only body mass and diet were also applied as an alternative in supplementary. All the data were analyzed together and visualized in 1-million-year time bins, as with the taxonomic clusters.

To test the consistency of the results obtained by PAM, we also conducted a non-metric multidimensional scaling (nMDS) analysis on the taxonomic and functional dissimilarity matrices using the *vegan* package[76]. NMDS is an ordination technique to visualize dissimilarities for non-Euclidean data which can visualize the communities across a low number of dimensions. The closer the two communities are in the plot, the more similar their composition is. Stress values indicating the goodness-of-fit were checked, with optimal values considered to be below 0.2[77].

Additionally, correspondence analyses of the community-genus and community-trait data matrices were conducted to visualize the relationships between communities, genera, and traits for providing visual information on which taxa and traits have the strongest influence on (i.e., most strongly define) the three continents or three clusters determined by the clustering analysis. Correspondence analysis is a multivariate statistical technique used for visualization of the relationships between multiple categorical variables.

**Reporting summary**

Further information on research design is available in the Nature Portfolio Reporting Summary linked to this article.

## Data availability

Data used to generate Figs. 1–4 can be found in Supplementary Data 1–3. The fossil data used in this study are provided in Supplementary Data 1. The trait data used in this study are provided in Supplementary Data 2. The synonym data are provided in Supplementary Data 3.

## Code availability

The R script used in this study is available at Zenodo: https://doi.org/10.5281/zenodo.18733763.

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

## Acknowledgements

This study was funded by the European Research Council through an Advanced Grant to I.d.l.T. (BICAEHFID, project number 832980). Additional support came from the Diversity Dynamics Department of the Museum für Naturkunde. We thank Haowen Tong, Shuwen Pei, Dong-dong Ma, Xin Ding, Jingjing Bie, Jiachen Cai and Carlos Fernandez for their assistance with the faunal records and related information, members from Amniota Lab at the Museum für Naturkunde and the BICAEHFID project for discussion.

## Author contributions

F.B. and I. de la T. conceptualized the study. J.S. and F.B. developed the methodology. J.S. and I. de la T. were responsible for data collection and curation. J.S. conducted the formal analysis and wrote the original draft of the manuscript. F.B. and I.de laT. reviewed and edited the manuscript.

Funding was acquired by I. de la T. All authors approved the final version of the manuscript.

## Funding

## Competing interests
The authors declare no competing interests.
