## [Transparent Peer Review file · Nature Communications]

No evidence that hominin dispersal across Eurasia was part of a wider turnover in mammal distributions

Corresponding Author: Mr Jijia Sun

Version 0:

Reviewer comments:

Reviewer #1

(Remarks to the Author)

Review of Eurasian-African Large Mammal Biogeography: Taxonomic interchange, functional stability, and Late Pleistocene restructuring by Sun et al.

The authors seek to test two hypotheses, namely that Homo was a part of a greater faunal exchange between Africa and Eurasia, and whether the arrival of hominins dramatically changed faunal communities. The authors test these hypotheses by examining community structure defined taxonomically and functionally. These are interesting hypotheses, which have the potential of providing insight into some major events in the evolution of our own species as well as the structure of mammal communities we find today. However, there are a number of points the authors will need to address to adequately test these hypotheses. Some may be surmountable, but others may not be achievable based on the quality of the data available. My major concerns are as follows:

1. The age of each site was calculated as the midpoint between the maximum and the minimum age reported. While this is a common way of dealing with chronological uncertainties for these large modelling efforts, it assumes a normal distribution of age probabilities, which is rarely warranted but mitigated somewhat by using 1 Ma time bins. What's more difficult to accept is that this logic is extended to geological intervals. In many sites examined by the authors, the estimated age of the site is significantly larger than the age bins used in subsequent analyses – at worst, the age range of, for example, Dadingshan, is between 5.333 and 0.0117 Ma, with the midpoint at ~2.7 Ma. There's no reason why this site belongs better in the 2-3 Ma bin than the 5-6, 4-5, 3-4, 1-2, or 0-1 Ma bins. The estimated age needs to be smaller than the bins examined. This means increasing the age bins, eliminating sites, or a combination of both.

2. How the functional groups were allocated and the taxonomic scale at which they were estimated is not, as the authors suggest, "relatively straightforward" (line 299). It's difficult enough at the species level for extant taxa, let alone at the genus level for fossil taxa. Assigning traits in a categorical fashion is probably the only way this can be meaningfully operationalised. However, these traits are not discrete but instead are a continuum, which becomes harder to justify at the genus level across three trophic guilds, and even more so for five locomotory categories for fossil species. For example, Ursus is classified as an omnivore. How is this defined and justified? In modern species, Ursus arctos has an omnivorous diet, but one that is dominated (>90%) by plants (e.g., Ogurtsov, S.S. The Diet of the Brown Bear (*Ursus arctos*) in the Central Forest Nature Reserve (West-European Russia), Based on Scat Analysis Data. *Biol Bull Russ Acad Sci* 45, 1039–1054 (2018). <https://doi.org/10.1134/S1062359018090145>). In contrast, Ursus maritimus is almost exclusively a carnivore. The authors could argue that the average diet of these two species means the genus as a whole is omnivorous. But this obscures some quite important ecological distinctions. This is exacerbated for locomotion – what distinguishes, for example, a scansorial versus arboreal species? Is the former one that spends a certain percentage of time on the ground? How then is that calculated for fossil species, some of which may only be known from cranio-dental fragments? At the very least, the authors need to provide detailed justifications for how these were assigned for each genus AND species, and where uncertainties (which will be many by virtue of the nature of the fossil record!) exist. These uncertainties, together with the loss of resolution resulting from examining the traits at the genus level, likely explain some of the strange results the authors get. For example, in the modern communities (Fig 1B), it appears that Bornean mammal communities cluster more closely with Arctic communities than they do with other Indonesian regions! This makes no sense to me and makes me question their fossil results (which as mentioned, will be based on worse data than modern taxa).

3. How do the authors define the continental landmasses? Africa is usually self-evident, but where do the authors draw the line between Europe and Asia? Is that even a meaningful distinction to make? They are both part of the same biological realm, meaning their species share a common phylogenetic history (Holt, B. G. et al. An update of Wallace's Zoogeographic regions of the world. *Science* 339, 74–78 (2013)). Perhaps a clearer definition of these, and what is meant by an "African

fauna” for example, would clear up some confusion I had reading through their results and discussion. It was not clear, at least to me, if when using the term African fauna the authors were referring to fauna from sites that are physically situated in that continental region, or sites that clustered most closely with other sites that were predominately in Africa. Put another way, how would the authors refer to a faunal assemblage from a site situated in Eurasia but dominated by African fauna? Other minor points to consider:

1. Line 149 – “literature describing individual taxa” – please provide this list
2. It would be useful for the authors to include some sensitivity tests, e.g. using different time bins, more limited functional groups (e.g. one trait, two traits), etc.
3. Figure 1 – I found this figure very hard to interpret. What are the genus clusters 1, 2, and 3? Likewise functional clusters? Also the colour scheme made it difficult to discern some of the patterns.
4. I think the chronofauna analysis needs to be extended. What is the mirror of this analysis doing (i.e. African site as anchor). Does selecting a site with known hominin influence on the fauna change these results (critical I think for really testing hypothesis 1).
5. Line 220 – you have gone to considerable effort to quantify everything, which will allow you to test whether these differences are statistically different, rather than just relying on the patterns in the figures.
6. Figure 4 – This figure could be clarified, consider colour choice and font sizes
7. Line 241 – to which figure or results are you referring to here?
8. Line 291 and 324-5 – would be worth reconsidering in light of my comments above. I would argue that, unless shown explicitly otherwise, this is the result of the nature of the fossil record and how the functional groups have been assigned.
9. Line 320 – you have the dataset in place to see if Homo is the only genus moving from Africa to Eurasia at this time. Is that the case?

Julien Louys

(Remarks on code availability)

Reviewer #2

(Remarks to the Author)
Response to the Authors

This article examines the evolution of taxonomic and functional structure in fossil and present-day large mammal communities from Africa, Europe, and Asia over the past 10 million years. Its objective is to assess whether the dispersal of hominins out of Africa approximately 2 million years ago was part of a broader wave of faunal migration into Eurasia and whether it was accompanied by significant changes in the functional structure of large mammal communities in Eurasia. The study’s results do not support the hypothesis that hominin dispersal was linked to a massive faunal migration from Africa. Instead, they indicate that the most relevant faunal exchanges during the Plio-Pleistocene occurred between Europe and Asia, while African faunas have remained taxonomically distinct from Eurasian ones for approximately 5 million years. Likewise, no evidence was found that the arrival of hominins in Eurasia was associated with significant modifications in the functional structure of large mammal communities.

Although the study presents an interesting and potentially relevant approach to understanding hominin dispersal, I believe there are methodological and interpretative issues that should be addressed in greater depth.

Title Recommendation

It would be advisable for the title to more explicitly refer to the ultimate objective of the study, which is to evaluate whether the dispersal of hominins out of Africa approximately 2 million years ago was part of a broader faunal migration and whether it altered the functional structure of large mammal communities in Eurasia.

Materials and Methods

The study aims to detect changes in the taxonomic composition and functional structure of mammal communities over time. To achieve this, it analyzes the taxonomic and functional similarity of both fossil and present-day communities across time and space. Although previous studies have addressed specific aspects of the evolution and distribution of certain mammal groups in these regions, the combination of taxonomic and functional analyses within a broad temporal and spatial framework represents a valuable and complementary contribution to the existing literature.

Choice of Functional Analysis Method

The functional analysis is based on transforming a dissimilarity matrix using PCoA before applying k-means. However, since the main objective is to identify clear temporal transitions in functional structure, it would be more intuitive to use the number of species in each functional category (size classes, diet, and locomotion) as input variables. This approach would avoid the need to transform the data into an artificial Euclidean space, as functional categories naturally define a Euclidean space. Additionally, it would allow for a more direct identification of transitions in functional composition, facilitating the analysis of changes across key periods.

If the goal were to study more subtle relationships between communities or work with more complex taxonomic or functional distances, the use of dissimilarity matrices and PCoA would make sense, as they capture more nuanced differences between communities. However, if the intention is to detect clear transitions in functional or taxonomic structure over time,

the method based on the number of species per functional category appears more efficient and straightforward. If the current approach was chosen for specific reasons, it would be advisable to explicitly state in the manuscript why this option is preferable to the direct use of functional categories.

Justification for the Choice of $k=3$ in Clustering

In the manuscript, the selection of $k=3$ is justified as the minimum number of groups necessary to distinguish the three continents. However, this choice does not appear to be supported by a statistical evaluation of the optimal data structure. Such a low number of groups means that each cluster encompasses significant functional and taxonomic diversity, potentially obscuring important ecological transitions.

Unlike, for example, the Great American Biotic Interchange (GABI), where the formation of the Isthmus of Panama created a land connection that structured dispersal patterns between North and South America, the dispersal of hominins out of Africa cannot be explained by the mere disappearance of a well-defined biogeographic barrier. In this case, if a reorganization of mammal communities occurred, it must have been the result of ecological processes. As a result, changes in taxonomic composition and functional structure of communities should not necessarily manifest at a continental scale but rather in smaller ecological units.

Given that the Euclidean space defined by the approximately 12,000 analyzed communities is extremely large, such a low number of clusters may have prevented the detection of significant transitions in community structure across different time periods. To assess whether the optimal number of clusters has been correctly determined, it would be advisable to explore standard statistical methods such as the elbow method or the silhouette coefficient, which would allow verifying whether $k=3$ is indeed the most suitable structure or whether a higher number of groups would better reflect the variability in community composition. If the data indicated, for example, that $k=4$ is the optimal number of clusters, forcing $k=3$ could lead to the merging of two distinct groups, making it harder to detect transitions between them.

Additionally, to assess whether there are discrete functional structures within the data, as expected according to Mendoza and Araujo (2022) and González-Trujillo et al. (2024), the application of AMD analysis could be considered (see Supporting information, Mendoza and Araujo, 2022).

If there is a solid justification for the choice of $k=3$, it would be advisable to explain it in greater detail in the methodological section, substantiating its validity against other alternatives that might offer a more precise resolution and better capture the dynamics of change in mammal communities.

Interpretation of Statistical Analyses

The article employs advanced statistical analysis methodologies, which make it valuable for paleobiogeography. However, it would be beneficial to include a brief, accessible explanation of the function of each analysis and the interpretation of its results. Although the article is intended for a specialized audience, a more intuitive description of each methodological approach would facilitate understanding for readers unfamiliar with these techniques.

Taxonomic and Functional Dissimilarity: Missing Key Information

The manuscript states that total dissimilarity consists of two distinct components: replacement and nestedness. However, it does not explain how these components were incorporated into the analyses or how they are interpreted in the results. Since these two components capture different dimensions of dissimilarity between communities, their treatment in the study may significantly influence the interpretation of the observed patterns.

It would be helpful for the authors to clarify whether both components were analyzed separately or if only total dissimilarity was considered. Additionally, it would be relevant to specify whether they play different roles in identifying taxonomic and functional clusters, as well as whether their relative weight has specific biogeographic implications in different temporal or spatial contexts.

Furthermore, the manuscript states that the dissimilarity matrix is calculated within each 1-million-year interval and that fossil data are visualized in these intervals. Since the k -means analysis is performed with $k=3$, can the authors confirm whether clustering is applied independently within each time interval, or whether clusters are tracked across intervals?

This would imply that cluster structure could change between periods, as community composition varies over time, and that in each interval, communities are assigned to the three clusters solely based on the dissimilarity structure at that moment. If the goal is to assess the persistence of clusters over time, have the authors considered analyzing to what extent the groups formed in one interval remain in the following ones? Currently, the manuscript does not explicitly mention whether clusters are compared between periods or if they are simply analyzed independently in each interval.

A clarification on this point would be appreciated to better understand the methodology used.

Structure of the Materials and Methods Section

Currently, the structure of the Materials and Methods section does not clearly reflect the relationship between the four methodological approaches used in the study (taxonomic clustering, functional analysis, chronofaunas, and nMDS). The inclusion of specific subsections for each of these approaches, organized coherently and with direct references to the figures presenting their results, would significantly improve the clarity of the study design.

Greater Clarity in the Description of Analyses

Providing more details on how each method is applied and its specific role in the analysis would help improve clarity and facilitate the interpretation of results. In particular, it would be useful to explain why each method is employed, what specific problem it seeks to address, and what type of information is obtained at each step of the procedure. This would better contextualize the results within the study's objectives and provide a more precise interpretation of the observed patterns.

Results

Structure of the Section

As in the Materials and Methods section, the structure of the Results section does not clearly reflect the relationship between the four methodological approaches used in the study. Including specific subsections for each of them, organized coherently, would significantly enhance the clarity of this section.

Greater Depth in the Analysis and Interpretation of Results

The description of the results is often too brief, making it difficult to follow and understand. It would be advisable to provide more detail on the findings, ensuring that the presentation of the data is clear and complete before their interpretation in the Discussion section.

Clarity in Figure Captions

The figure captions in the manuscript are too general and do not adequately differentiate the various panels within each figure. This hinders the interpretation of the results. It is recommended to expand the descriptions to include more detailed and specific information about each panel.

Interpretation of Results and Consistency with Figures

In the Results section, some statements do not seem to be fully supported by the data displayed in the figures. Below are specific inconsistencies along with suggested reformulations:

- Line 191. The statement: "The three taxonomic clusters generally corresponded to a Miocene African-Eurasian fauna, a Miocene-Pleistocene African fauna, and a Pliocene-Pleistocene Eurasian fauna (Fig. 1)" does not seem to reflect the spatial patterns observed in the maps. A more precise reformulation could be:
"The three taxonomic clusters generally corresponded to a predominantly Eurasian fauna in the Late Miocene (Cluster 1), which is now found in North Africa, the Middle East, and South Asia; a fauna that was present on all three continents until the Early Pliocene, when it became restricted to Africa, and is currently found in sub-Saharan Africa (Cluster 2); and a fauna that emerged in Europe in the Late Miocene, gradually expanded into Asia, and currently dominates Europe, northern Asia, and parts of eastern Asia (Fig. 1A)."
- Line 193. The statement: "From 10 Ma to 6 Ma, the African and Eurasian sites shared a very similar faunal composition..." does not appear to be supported by Figure 1A, where the Eurasian cluster (Cluster 1) is present in Europe and Asia but is almost absent in Africa.
- Line 197. The statement: "This second cluster became dominant in Africa from 5 Ma until today, distinguishing African faunas from those in Eurasia since that time." does not align with the distribution observed in Figure 1A. To claim that a cluster became dominant, one would expect a more widespread distribution across the continent, which does not seem to be reflected in the data shown. A more precise reformulation could be:
"This second cluster is present in all available sites in Africa from 5 Ma until today, distinguishing African faunas from those in Eurasia since that time."
- Line 202. It is mentioned that between 6–5 Ma and 4–3 Ma, Asian faunas still belonged to the "Miocene" cluster (Cluster 1). However, during this period, there are few Asian localities with sufficient data, and these exhibit all three cluster types. This suggests that the statement should be nuanced to better reflect the observed distribution.
- Line 253. The claim that Eurasian and African faunas showed high similarity between 10 and 7 Ma is not consistent with what is observed in the maps. In Figure 1A, the Eurasian cluster (Cluster 1) is well represented in Europe and Asia but has little presence in Africa.
- Line 255. The idea that a distinctive African fauna emerged between 7 and 6 Ma should be nuanced, as Cluster 2 was already present in earlier periods. Rather than an abrupt change, it seems to be a gradual process in which the differences between continents became more pronounced over time.
- Line 256. It is mentioned that between 6 and 5 Ma, a new cluster appeared in northwestern Europe (Cluster 3), which replaced more archaic Asian faunas during the Pliocene. However, the maps show that the previous clusters continued to coexist for some time in different regions, suggesting that the transition was more gradual than the text implies. It is recommended to reformulate this section to more accurately reflect the spatial distribution of clusters in each period and avoid overly simplistic interpretations of faunal composition changes.

Conclusion

Overall, this study addresses a relevant question about the relationship between hominin dispersal and the dynamics of mammal communities over time. However, several methodological and interpretative aspects require further development and justification. The selection of the number of clusters without clear statistical support, the brief methodological explanations, and the lack of correspondence between some results and the figures affect the robustness of the conclusions. To improve the clarity and rigor of the study, it would be advisable to include more detailed justifications for the analytical decisions made, restructure the presentation of results to more accurately reflect the observed patterns, and expand the discussion on the biogeographical and ecological implications of the findings. Addressing these aspects would significantly strengthen the contribution of the article and facilitate its interpretation within the broader framework of mammal community evolution and hominin dispersal.

I hope my feedback contributes to strengthening the manuscript.

Best regards,
Manuel Mendoza

(Remarks on code availability)

Reviewer #3

(Remarks to the Author)

I have read the study by Sun et al. with great interest as the topic of hominin co-migration with other fauna and their potential influence on the functional composition of large mammal communities is of central importance to understand the onset and extent of hominin impact on global ecosystems. As I am not an expert in the methodology used in the study, I cannot provide fair judgment on the quality of the applied methods. However, testing of the two main hypotheses outlined in the paper (hominin-mammal co-migration and hominin impact on mammalian functional composition) is mostly studied by examining fossil evidence from non-hominin mammals and it thus seems that the proposed hypotheses are tested tangentially at best. Given the focus of the proposed hypotheses on hominin-related dynamics, a more direct analysis of the relationship between hominin fossils and other mammalian fossil assemblages is needed. For example, as some fossil assemblages contain fossils from the Homo genus, it would be instructive to test if this Homo presence in any way affected the taxonomical and functional composition of the rest of the assemblage (compared to assemblages that do not contain Homo remains). While the global trends indicate that large taxonomical and functional changes occurred either before 2 mya or during Late Pleistocene, this does not exclude the possibility of hominin impact at a more local scale (testable by comparing Homo-containing and non-containing assemblages, as mentioned above). Additionally, to provide a stronger case for Homo not affecting mammalian biogeography, climate or similar data that have been reconstructed for the studied period could have been used to test if environmental shifts drove the observed patterns.

In general, I would like more elaboration on the selected methodology and data selection. Specifically, I agree that body mass and diet likely capture the most fundamental functional roles of taxa in a community, but I am unsure how locomotion would be informative (especially as it is defined with respect to habitat type, which may or may not reflect locomotion as the ability to move from one place to another). Additionally, all life history traits are correlated and I am uncertain whether the methodology used adequately accounts for the strong correlations among these traits and their potential impact on the observed patterns. Generally, the methodology needs further justification to ensure it is more self-contained and the choices made are clearly explained. More specific details follow below.

Line-by-line comments

Line 103: Cantalapidra should be Cantalapedra

Line 134: Current IUCN ranges are heavily influenced by human impact - therefore it should be addressed how this may impact the analysis. Additionally, mammalian present natural ranges from the PHYLACINE database (10.1002/ecy.2443) may be more appropriate to use.

Line 153: More detail is needed about how the "pairwise distance matrix" is calculated, rather than a simple reference to an R package.

Line 158: "k=3" needs to be justified by a plot of variance explained vs. the number of clusters, a simple "we found" statement is insufficient

Lines 161-168: A single reference site is chosen as a point of reference. However, I suggest multiple similar points should be tested to show robustness of the observed patterns.

Lines 169-170: The "genus-trait matrix" and the "Gower dissimilarity matrix" need to be defined.

Lines 175-176: Why was the same number of clusters used in functional analysis as in the taxonomic analysis? The functional traits contain many categories and thus many more combinations of traits are possible than 3. I therefore do not see how "following the same principal of the taxonomic clustering" is justified.

Line 200-201: The sentence "Homo was present in 22 out of 162 sites." seems awkward in this position as it is not followed up with a reason why this might be an important statistic to report.

Line 213: "difference peaking"? - should it not be "similarity peaking"? Additionally, the peak is only clear for Asia, after which it drops off towards the Late Pleistocene (an explanation for this trend might be good to include), while the similarity reaches a plateau for Europe in the Mid to Late Pleistocene. Additionally, an explanation of the lines and shaded area would be needed to better understand Figure 2.

Line 220: The separation between geographic clusters (Fig. 3A) is described as strong. I think this needs to be attenuated as clustering is not very clear, especially during the Miocene.

Line 241-246: The paragraph argues that extant taxonomic and functional clustering "followed a similar latitudinal pattern", which is not very accurate given the maps in Fig. 1A,B.

(Remarks on code availability)

Reviewer #4

(Remarks to the Author)

Thank you so much for asking me to review this manuscript, which I approached with great interest and enthusiasm! I am happy to disclose my identity to the authors, and if there are any questions about my comments in this review, I would welcome direct communication.

This study is generally great, and I would like to see it published in some form. The data collection alone must have been a feat; despite the increasing availability of fossil data through public databases such as NOW and the PBD, these data are not easy to merge and process. Additionally, I note that new Asian site data is included herein – 92 new previously unpublished sites, if I have understood it...? It's not clear how these data were collected or acquired, but I would like to suggest that this individual(s) is included in the author list. The compilation of new data is a major contribution to science and if the data hasn't previously been published elsewhere then I recommend that the data contributor join the author list.

The major comment I wish to start with is that this manuscript contains a large-scale study of major shifts (or not, as the case may be) in the taxonomic composition and functional structure of mammalian communities over ten million years, in one-million-year intervals. The time scale of this may be at odds with the scale required of a study of hominin migrations out of Africa. This is not a fatal flaw in my opinion! But, somehow this paper needs to be framed in a different way. The introduction details two major phases of hominin dispersal out of Africa, emphasizing the first one around 2 mya. The hypotheses are both testing issues relating to these dispersals. This gives the impression that the study will focus on the time periods relevant to these dispersals and although they are incorporated into the study, they are not the analytical focus. Therefore, my advice would be to reframe – the intro should be more 'honest' about the ensuing analysis, centering mammal communities and their changes over long period of time, rather than hominin dispersals from 2 million years onwards. But I appreciate that hominins may be thought of as a greater point of interest by some – not me, as in my opinion a study of mammal communities over 10 million years (during which hominins happened to disperse out of Africa) is in and of itself an excellent study! But if the authors did wish to truly make the focus of the entire paper about hominin dispersals and their (potential) correlation with changes in mammal communities, I would say that the earliest time periods could be removed from the study as they aren't pertinent, and the analyses should look at a much finer grained time scale to better capture the time when hominins disperse and arrive in new places (and indeed enough time goes by to see any changes that might occur in the communities as a result). In my opinion this would take far more work than reframing the study to focus on mammal communities over time, painting their structure and function with a broad brush to look at macro patterns. Within that framework the authors can comment on events during that long timeframe that have the potential to modify communities including, but not limited to, hominin dispersals.

Below I outline additional comments and questions associated with relevant line numbers. Since I have suggested reworking the framework and hypotheses, it is difficult to direct some of my comments at a hypothetical newly framed paper. Therefore, it is probably easier if I provide my comments in the order they came to me whilst reading the manuscript. I have five substantive questions about the methods because I wasn't clear on what all of the analyses accomplished in terms of testing the two stated hypotheses. I don't make many comments about the discussion section because the paper's reframing would necessitate a new discussion, and the thoughts that came to mind were more or less the same as those that arose when I read the introduction (about the timescale and granularity of the study), and which are described above.

Line 33 or thereabouts – you could acknowledge the newly reported 1.95 Mya Romanian material by Curran et al 2025 (<https://doi.org/10.1038/s41467-025-56154-9>)

Lines 47-49 – this paragraph is only a single sentence describing the 2nd OoA event, which isn't consistent with the summary of the 1st event. Could more be added for better context?

Lines 50-63 – Could this paragraph be rewritten so that it reads more smoothly? I also recommend that you delete the word natural (palaeoenvironments are of course natural, so that word is redundant) and delete the word huge (which is a bit colloquial).

Lines 64-83 – the first part of this paragraph refers to herbivores and then carnivores, so I suggest you stick with this order throughout the paragraph and review the herbivore info first, then move onto carnivores. You could then more easily link to the next paragraph (lines 84-88), which stands a bit awkwardly on its own (or better yet, incorporate it).

Lines 99-115 – this paragraph mostly consists of paper-by-paper one sentence summaries, rather than a synthesis. Also, and with the caveat that this is a stylistic point and perhaps the authors don't find fault with it, but nearly each sentence begins with the name of an author(s). The text shouldn't be about the people, it should be about their science. This style may be what has got you stuck in paper-by-paper summaries, as well. I would recommend that this paragraph is rewritten so that it doesn't name other researchers throughout (or that this practice is reduced) and the material is better integrated.

Lines 116-122 - Could this paragraph be linked more clearly with the above paragraph(s)?

Line 134 – geographic distribution data comes from IUCN and although this would work for extant taxa in quite recent history, I'm not sure it works well for taxa in the deeper past, even if they are extant. For example, Thailand has had a

number of local extinctions during the Pleistocene, but many of these taxa live elsewhere in Asia today (pandas and orangutan). So, the IUCN data won't provide the correct distribution data for these species. This is my first major query about the methods in this paper. I'd like to know how this issue was accounted for or if it could be (or evidence to support that there is no need to).

Lines 145-147 – are these the same diet and locomotion categories used by others, so you are following an established system?

Line 149-150 – It would be more succinct to simply state what the analysed body categories were rather than explain that small mammals were excluded after their body size categories are given earlier in the paragraph.

Lines 158-160 – how did you find that $k=3$ is the minimum number of clusters needed to distinguish the continents? That rather makes intuitive sense since there are three continents, but the evidence to support this statement should be provided. Is it in the supplementary info, but I missed it? This is my second major methods-related question. Also, the supplementary info should be referred to wherever it includes data not summarized/presented in the main text. There was a lot of missing info such as the names of the sites, their constituent fauna, their geological dates, their references, and info about the trait assignments for each genus, and those references. There are files pertinent to these data available to download, but they should be referred to in the text.

Line 177-178 – Add a statement of what nMDS was used for at the end of the sentence, like “in order to...” which circles back to the stated hypotheses. It generally feels like the methods should all be more explicit in how each analysis ties to the stated hypotheses. It was hard to tell how each test contributed to the whole. This is my third major comment on the methods.

Lines 191-207 – figure 1 is referred to, but it has two parts, A and B, which need to be referred to where relevant.

Line 208 – Figure 1 itself should have the colour legend defined in it, and also in the caption that goes with it. Also, for the Present, it looks like you included areas of very high latitude where you don't have fossil sites represented. Have I read that correctly? Would it make sense to have a latitudinal cut-off for the Present that matches where the fossil sites are?

Lines 210-216 – I must confess that I am not that familiar with chronofauna analysis and that I struggled to get my head around the purpose of it here. I have to ask if it is necessary to the overall story. In looking at the results in figure 2 it looks like the Asian and European fauna follow the exact same curve and that they remain the same distance apart until just before 2.5 million years ago when they diverge from each other at the start of the Pleistocene. So, they both grow increasingly dissimilar to SSMZ until their curves diverge and Asia starts to look more similar to it. This isn't commented on, but it merits an explanation, particularly the timing. This is my fourth major comment on the methods.

Lines 230-238 – it wasn't until these results of the correspondence analyses that I realised I was unclear about how the time variable was considered in them. I don't think it was accounted for, am I right in this? If that is so, everything has been lumped together over a 10-million-year period...? To me, that doesn't make sense. Firstly, it is known that taxonomically these three continents differ from each other, but that Europe and Asia share more similarities. So, this doesn't seem entirely new, but a confirmation of what is known - but it would be insightful if the differences were explored over time to look for more nuanced trends (although I suspect there would be a high level of differentiation throughout the time sequence). Secondly, my feeling is that lumping the trait data together for 10 million years was likely to result in a lack of differentiation between the three continents, and again there would be more nuance and insights if the data were analysed by time bin. This is my fifth major methods comment.

Line 290 – these faunal geographic regions are not denoted in figure 1A where the reader is referred to, and have not been mentioned elsewhere. It's true that some readers will be familiar with them, but some sort of explanatory statement is necessary.

Line 298 – “Data inaccuracy is unlikely...” That is a strong statement! There are surely a host of inaccuracies in the data which have not been acknowledged (which are not the authors' fault, but simply the reality of working with fossil fauna community data)... taphonomy, depositional context, difficulties with genus identification, sampling etc. Aspects of the methods could also confound the analyses – for example, the dietary categories are blunt, and include the category 'omnivore' which is something of a catch-all, and sites with only five unique genera were included, which is a very small number. Other researchers have grappled with this latter issue, Louys et al. 2009 finding that 12 taxa was an acceptable minimum number (<https://doi.org/10.1016/j.jas.2009.06.012>).

Finally, the Discussion seems somewhat under-referenced, with only about 8 citations. I appreciate that there is a suggested limit to the number of references in this journal, but wonder what can be done to make sure that the relevant body of literature is adequately acknowledged?

Again, I reiterate my belief that this is a wonderful study! Although some aspects need to be ironed out and I have many questions and suggestions, these are offered as constructive pieces of advice. I hope that they do prove helpful in the revision process, and I am truly looking forward to seeing this work in publication.

Kris (Fire) Kovarovic
Durham University, UK

(Remarks on code availability)

The R code is available but I am not a competent R user so I have not reviewed it.

Version 1:

Reviewer comments:

Reviewer #1

(Remarks to the Author)

My major concerns have been addressed by the authors, who have provided additional data and refined their analyses. No further comments from me.

(Remarks on code availability)

Reviewer #2

(Remarks to the Author)

Overall, my concerns have been satisfactorily addressed and the revisions improve the manuscript. Three essential adjustments remain: (1) correct the definition of AMD to Average Membership Degree throughout the manuscript, including the relevant figure caption(s) (e.g., Supplementary Fig. 1) and all supplementary text; and (2) remove any statement that $k = 3$ was chosen "because there are three continents" and replace it with the data-driven justification (silhouette and AMD). If these minor points are addressed, I am happy to recommend the manuscript for publication in Nature Communications.

(Remarks on code availability)

Reviewer #3

(Remarks to the Author)

The manuscript has improved considerably. However, some minor revisions are still necessary. For example, the manuscript should be structured so that figures are referenced in the correct order (e.g., at present, Fig. 4 is cited after Fig. 1 but before Figs. 2 and 3). In addition, the language requires further refinement. I recommend that the manuscript be thoroughly revised by a native English speaker. Additionally, I recommend providing the original code used to analyze the data and generate the figures, if it has not already been supplied.

(Remarks on code availability)

No code detected

Reviewer #4

(Remarks to the Author)

Thank you for the opportunity to review this manuscript in its revised form. I appreciated being able to also read the other three reviewers' comments, through which it became evident where we had similar suggestions or points of confusion.

In my second round of comments below, I have copied and pasted my initial points with the authors' replies, and I respond directly to each point here (I didn't do this where I had no further comment - only for issues on which I would like to follow up). Hopefully, this makes it easier to follow the review process as a conversation, and I am more than happy to continue this conversation as necessary. After that, I have added new comments with line numbers referring to the revised manuscript.

I can see where meaningful changes have been made to the methods and the processing/treatment of the data (randomisation was a great idea, there is a sensible approach now to putting taxa in the time bins, amongst other improvements). But, the manuscript itself does not make these improvements clear to a reader. The responses made to the reviewers provide a much better explanation of the changes, and why they were implemented, than the manuscript which is not well-written in places (grammatical mistakes and the use of colloquial language – "tiny" "basically") and does not synthesise the background information or the results for an impactful discussion. It needs to be heavily edited so that the writing, structure and clarity are all improved. The main text of the manuscript is at the 5000-word limit, and I can certainly appreciate the challenges of packing so many analyses into that limit, but to achieve the quality of manuscript that I would expect in Nature Communications, there is much work to be done.

My other major comment is that you have decided not to undertake the reframing of the paper as I suggested and, unfortunately, I don't think it is helping you "sell" this study very well. Some of my comments below will attest to this. The broad categorisations of fauna that you have used and the ten-million-year time period covered do not support the aims of your paper as they are currently described. Some of the results are not new or surprising, and the results specifically from the relevant hominin timeframe should be explored with much greater depth. Please consider how to appropriately and effectively introduce the main themes of this paper, and link them clearly to the datasets analysed. There is a disjunct between them at present.

FOLLOW UP ON INITIAL POINTS AND RESPONSES:

This study is generally great, and I would like to see it published in some form. The data collection alone must have been a feat; despite the increasing availability of fossil data through public databases such as NOW and the PBD, these data are not easy to merge and process. Additionally, I note that new Asian site data is included herein – 92 new previously unpublished sites, if I have understood it...? It's not clear how these data were collected or acquired, but I would like to suggest that this individual(s) is included in the author list. The compilation of new data is a major contribution to science and if the data hasn't previously been published elsewhere then I recommend that the data contributor join the author list. All the data newly compiled for the study were collected by the lead author from the literature. The raw data itself has been made available as a supplementary file "fossil.csv", and will form part of the BICAHEGIS database, which will be made public shortly.

If the 92 Asian sites referred to as 'new' were in the literature and available for compilation by the author, then they are not in fact new. They are published works. This description needs to be changed in the text. They are simply 92 sites that form part of the Asian site dataset. The same goes for the description of the Romanian site that was published after the initial submission and then added to the analyses that form the most recent version of the manuscript.

The major comment I wish to start with is that this manuscript contains a large-scale study of major shifts (or not, as the case may be) in the taxonomic composition and functional structure of mammalian communities over ten million years, in one-million-year intervals. The time scale of this may be at odds with the scale required of a study of hominin migrations out of Africa. This is not a fatal flaw in my opinion! But, somehow this paper needs to be framed in a different way. The introduction details two major phases of hominin dispersal out of Africa, emphasizing the first one around 2 mya. The hypotheses are both testing issues relating to these dispersals. This gives the impression that the study will focus on the time periods relevant to these dispersals and although they are incorporated into the study, they are not the analytical focus. Therefore, my advice would be to reframe – the intro should be more 'honest' about the ensuing analysis, centering mammal communities and their changes over long period of time, rather than hominin dispersals from 2 million years onwards. But I appreciate that hominins may be thought of as a greater point of interest by some – not me, as in my opinion a study of mammal communities over 10 million years (during which hominins happened to disperse out of Africa) is in and of itself an excellent study! But if the authors did wish to truly make the focus of the entire paper about hominin dispersals and their (potential) correlation with changes in mammal communities, I would say that the earliest time periods could be removed from the study as they aren't pertinent, and the analyses should look at a much finer grained time scale to better capture the time when hominins disperse and arrive in new places (and indeed enough time goes by to see any changes that might occur in the communities as a result). In my opinion this would take far more work than reframing the study to focus on mammal communities over time, painting their structure and function with a broad brush to look at macro patterns. Within that framework the authors can comment on events during that long timeframe that have the potential to modify communities including, but not limited to, hominin dispersals.

Many thanks for the advice. In this study, we aim to explore the driving factors and influences on the large mammal communities and hominins out of Africa. You are correct that a finer-scale study (in terms of temporal and maybe also geographic coverage) could have addressed the question at higher resolution. However, we intentionally designed our study at the coarse scale here (our choice of genus level, and the three traits also reflects this). The time scale is set to 10 Ma so that we can have a better and larger view of the whole large mammal community over Eurasia and Africa. As you point out, the objective need not have focused on hominins, but in this case this does in fact honestly reflect the objective of the study from the beginning (part of the larger BICAHEFID Project). Reframing the intro part would therefore result in a loss of focus on hominins dispersals, which is an important part of this study.

Most readers will not know what the larger BICAHEFID project is and that it focuses on Homo dispersals (I had to Google it myself), so that rationale will not support the current structure or narrative of the manuscript. If your intent was to focus entirely on human dispersals, there were better ways to analyse the data that would have allowed you to identify more nuanced differences in the communities over the shorter timeframe that applies to these dispersals rather than a 10-million-year view. This was recognized by other reviewers; the broad brush of the dietary categorisations is one such example, as is the use of genera rather than species. There are very good reasons for your choices, but they are particularly good choices for studying a long time period rather than a shorter timeframe like the one that relates to hominin dispersal.

Line 33 or thereabouts – you could acknowledge the newly reported 1.95 Mya Romanian material by Curran et al 2025 (<https://doi.org/10.1038/s41467-025-56154-9>)

We added it in the texts and also revised the faunal data in our dataset.

Great, it was serendipitous that this was published so you could include it! However, you properly integrate it into your introduction, where it is mentioned in the last sentence of the first paragraph -this is out of place in the chronology you described which begins at 1.8 Ma.

Lines 47-49 – this paragraph is only a single sentence describing the 2nd OoA event, which isn't consistent with the summary of the 1st event. Could more be added for better context?

Here we concluded the driving factors of the dispersal of Homo sapiens. To keep it clear and concise, we would like to keep it this way.

I'm afraid that it doesn't read well like this, and you need to condense and streamline the intro. Readers can't easily follow which Homo dispersal event or events you are going to focus on. You have not provided the level of detail necessary to justify discussing the second dispersal here.

Lines 64-83 – the first part of this paragraph refers to herbivores and then carnivores, so I suggest you stick with this order

throughout the paragraph and review the herbivore info first, then move onto carnivores. You could then more easily link to the next paragraph (lines 84-88), which stands a bit awkwardly on its own (or better yet, incorporate it).

Here we didn't quite understand the reviewer's suggestion. The order is first all mammals, then herbivores, then carnivores, followed in the next paragraph by further discussion of carnivores.

Apologies for not being clear. You have a long paragraph covering herbivores and carnivores, followed by a two-sentence paragraph relating to carnivores that does not make sense standing on its own. You need to rewrite this part of the intro, so it is smoother.

Lines 99-115 – this paragraph mostly consists of paper-by-paper one sentence summaries, rather than a synthesis. Also, and with the caveat that this is a stylistic point and perhaps the authors don't find fault with it, but nearly each sentence begins with the name of an author(s). The text shouldn't be about the people, it should be about their science. This style may be what has got you stuck in paper-by-paper summaries, as well. I would recommend that this paragraph is rewritten so that it doesn't name other researchers throughout (or that this practice is reduced) and the material is better integrated.

The paragraph has been rewritten.

It reads much better but 1) it could still synthesise the information more concisely and 2) you have added a new paragraph above it (lines 93-102 in the revised draft) that follows the same practice of sentences one after the other naming and summarising a single paper. This style is not working well for what you need to do in the introduction. At the moment, your introduction is a bit more than two pages. By contrast, the Discussion is just under two pages.

Lines 116-122 - Could this paragraph be linked more clearly with the above paragraph(s)?

This is the paragraph for introducing the hypotheses used in this study.

I understand the point of the paragraph, but my comment was about how it links to the previous text. You need to draw a connection between the intro/background and the hypotheses you want to test more explicitly.

Line 134 – geographic distribution data comes from IUCN and although this would work for extant taxa in quite recent history, I'm not sure it works well for taxa in the deeper past, even if they are extant. For example, Thailand has had a number of local extinctions during the Pleistocene, but many of these taxa live elsewhere in Asia today (pandas and orangutan). So, the IUCN data won't provide the correct distribution data for these species. This is my first major query about the methods in this paper. I'd like to know how this issue was accounted for or if it could be (or evidence to support that there is no need to).

As noted in response to a previous reviewer's comment, in our revision we also conducted our analyses on the PHYLACINE dataset, which includes both 'current' and 'present natural' datasets. The latter estimates geographic ranges in the absence of human impacts. Results of the PHYLACINE analyses are similar to those using IUCN data (shown as Supplementary Fig. 4)

This is a great way to address the issue I highlighted. However, you have not explained why you use the PHYLACINE dataset in the paper. Explain the potential problem with using IUCN data, and what the PHYLACINE data is based on that helps you get around the IUCN problem.

Lines 145-147 – are these the same diet and locomotion categories used by others, so you are following an established system?

Yes. The sources of the trait scores are listed in the same paragraph. We also added the reference in the supplementary spreadsheet file "trait.csv"

My apologies, I didn't quite describe my question very well! You do note the sources of your diet and locomotion/spatial categories and say that your trait data are "based on" them (lines 149-151). Are they FROM these sources or BASED ON them? This is an important distinction. If they are based on them, how did you make decisions about which categorization systems to use, or how to reconcile them from different sources? What did you do to ensure that you were consistent in your categorisations when the data came from multiple sources?

Line 149-150 – It would be more succinct to simply state what the analysed body categories were rather than explain that small mammals were excluded after their body size categories are given earlier in the paragraph.

For genera that span more than one size category, the exact value is determined by randomly choosing a single value early in the analysis. The exclusion of the body mass categories 1 and 2 therefore needed to be done after the randomized choice of the body mass (all shown in the R script provided).

Ok, that makes sense, but it needs to be explained in the main text, or you need to find a way to avoid a reader asking the question, as I did. You can't assume that readers will look over the R code to learn the answer.

Lines 158-160 – how did you find that $k=3$ is the minimum number of clusters needed to distinguish the continents? That rather makes intuitive sense since there are three continents, but the evidence to support this statement should be provided. Is it in the supplementary info, but I missed it? This is my second major methods-related question. Also, the supplementary info should be referred to wherever it includes data not summarized/presented in the main text. There was a lot of missing info such as the names of the sites, their constituent fauna, their geological dates, their references, and info about the trait assignments for each genus, and those references. There are files pertinent to these data available to download, but they should be referred to in the text.

We now conduct a silhouette coefficient analysis for determining the value of k . The best value of k for fossil data is 2 and 3 respectively for taxonomic and functional clusters. The best value for extant data is higher. But our aim was to compare the differences between the extant and fossil data, so we determined that all values of k should be 3. We provided more plots in the Supplementary as s supplementary material. Additionally, we added the reference for the supplementary material in the main text.

This is an interesting approach, but I am still not following the logic of selecting $k=3$. Please bear with me! The silhouette coefficient analysis results say that the best value of k is 2 for fossil functional clusters, and the AMD results in 2 for fossil taxonomic clusters.

The extant clusters do not seem to yield higher values of k as you say above; it is 3 in all cases, except for extant functional clusters which is 2. I am looking at Supplementary Fig 1 and this is what the caption describes. Three out of eight analyses say that $k=2$ and five out of eight say that $k=3$, so how do you then land on 3 being “best”? Neither the main text nor the supplementary figure explain this clearly.

Lines 210-216 – I must confess that I am not that familiar with chronofauna analysis and that I struggled to get my head around the purpose of it here. I have to ask if it is necessary to the overall story. In looking at the results in figure 2 it looks like the Asian and European fauna follow the exact same curve and that they remain the same distance apart until just before 2.5 million years ago when they diverge from each other at the start of the Pleistocene. So, they both grow increasingly dissimilar to SSMZ until their curves diverge and Asia starts to look more similar to it. This isn't commented on, but it merits an explanation, particularly the timing. This is my fourth major comment on the methods.

Good point. The chronofauna visualization is in reference to a single site, and for our broader continental-scale perspective it is not necessarily relevant to describe the differences of the geographic curves in detail. Furthermore, we have now added three further site comparisons. The goal of these visualizations is to provide further support to the distinctiveness of the African assemblages in relation to the Eurasian ones, and to show the closer similarity and turnover of European and Asian assemblages among each other.

Adding other reference sites helps make the point of the chronofauna analysis clearer, and addresses another reviewer's suggestion that the analysis be extended. In looking at the new results now, though, I wonder if they are actually complicating the point you want to make - certainly you need to be more explicit in describing what these results show in relation to your paper's goals. Many studies show that that African fauna is different from Eurasian fauna in the past and present, so what does this demonstration do to support your hypothesis testing (which is also specifically related to a much more specific time period)?

Lines 230-238 – it wasn't until these results of the correspondence analyses that I realised I was unclear about how the time variable was considered in them. I don't think it was accounted for, am I right in this? If that is so, everything has been lumped together over a 10-million-year period...? To me, that doesn't make sense. Firstly, it is known that taxonomically these three continents differ from each other, but that Europe and Asia share more similarities. So, this doesn't seem entirely new, but a confirmation of what is known - but it would be insightful if the differences were explored over time to look for more nuanced trends (although I suspect there would be a high level of differentiation throughout the time sequence).

Secondly, my feeling is that lumping the trait data together for 10 million years was likely to result in a lack of differentiation between the three continents, and again there would be more nuance and insights if the data were analysed by time bin. This is my fifth major methods comment.

Yes, the fossil data are analyzed together in the correspondence analysis. The reason for this is to provide visual information on which taxa and traits have the strongest influence on (i.e. most strongly define) the three clusters determined by the clustering analysis. Analyses by 1 myr time bin, whether for taxonomic or functional data, are not feasible at this scale of analysis as the data would be far too few. An analysis at this smaller timescale would also require using species-level taxonomy and a more refined set of traits might be able to do this (e.g. Faith et al. 2019

www.pnas.org/cgi/doi/10.1073/pnas.1909284116)

Ok, that explanation makes sense, but it is not present in the manuscript. The text relating to correspondence analysis in the methods section is overly general. You need to explain what you have described above – what is the purpose of this analysis in light of your study's goals?

Line 298 – “Data inaccuracy is unlikely...” That is a strong statement! There are surely a host of inaccuracies in the data which have not been acknowledged (which are not the authors' fault, but simply the reality of working with fossil fauna community data)... taphonomy, depositional context, difficulties with genus identification, sampling etc. Aspects of the methods could also confound the analyses – for example, the dietary categories are blunt, and include the category ‘omnivore’ which is something of a catch-all, and sites with only five unique genera were included, which is a very small number. Other researchers have grappled with this latter issue, Louys et al. 2009 finding that 12 taxa was an acceptable minimum number (<https://doi.org/10.1016/j.jas.2009.06.012>).

Our point was that assigning body mass, diet, and locomotion using our coarse categories was largely uncontroversial for the majority of the taxa. We reworded this part to try make this clearer: There are possibly two explanations for these differences: 1) The fossil datasets may be severely inaccurate or incomplete; 2) A fundamental rearrangement of functional groups appeared in the time between the fossil and modern dataset, i.e. during the Late Pleistocene and Holocene. We think it is unlikely that this pattern is the result of inaccurate data, particularly given that the attribution of such fundamental traits as size, diet, and locomotion to both fossil and extant genera was in mostly cases done with a relatively high degree of confidence. Also in Louys et al. 2009, their method and aim were quite different from our study. They determined a minimum number of species for distinguishing different habitats - which is not an objective of ours. As a comparison, Kaya et al. 2018 (<https://doi.org/10.1038/s41559-017-0414-1>) also used a minimum of five taxa identified to the genus level for their biogeographic similarity analyses

I would strongly disagree that the use of the term “omnivore” is uncontroversial. You do not define it in your paper, but point readers only to the sources of your trait data, which does not explain how your study has used the info in the sources (I noted this earlier – did they provide different info on some taxa, or use different trait classification systems to your own?). Omnivore is a catch-all term, and a poor one at that because it does not have a stable definition. Another reviewer gave a great example of this with respect to bears. By “omnivore”, do you simply mean any animal that will eat a combination of vegetation and animal matter? What combination of these resources is the threshold for inclusion in the “omnivore” category? Given how broad the categories “herbivore” and “carnivore” are, many mammals could be classified as

omnivorous (many herbivores eat invertebrates or eggs, for example and many carnivores do not hesitate to snack on wild fruit). Similarly, the locomotion categories mix both actual locomotion behaviours and the space occupied by a species, and these categories come with their own controversies (in your case, the use of the term amphibious was a bit curious – it's synonymous with the term semi-aquatic which is more commonly used, in my experience...?). These are all well-known conundrums, but they just be addressed. The use of the term omnivore may have an enormous impact on your clustering – so how is it justified? I personally try to avoid its use as I don't believe it is a meaningful classification, but if you want to use it, it needs to be clearly defined and justified. Saying that there was high confidence in the attribution of fossil taxa to categories, or that the process of categorisation is straightforward, comes across as naïve.

With respect to the number of genera at each site included in your analyses, I understand your point and see that using five is a cut-off was sensible. However, you do not cite the Kaya reference in the methods section or explain the rationale, both of which should be added.

NEW COMMENTS ON REVISED DRAFT:

Figure 1 – the legend still needs to be more explicit with respect to what clusters 1, 2 and 3 represent. Give them better/descriptive names in the legend. The caption has added some more detail in line with another reviewer's request, but all of the clusters were not described, only some of them.

Line 56 – use a more scientific word than “thrived”

Line 133 – readers should be referred to the table where the published literature/references can be found.

Line 139 – grid cells are not the same thing as communities, so either use the term grid cell or explain that the word community used here refers to each cell

Line 146 – groups, not levels

Line 155-157 – where can readers find these analyses and results?

Line 165 – delete “in this study”

Line 166 – a list of three matrices?

Line 173 – refer readers to the relevant figure/supplementary section

Line 241-242 – older and younger, not old and new

Line 243 – dominated not -ing

Line 290 – rephrase “started to become low again”

Line 302-303 Figure 3 caption – this sentence isn't clear. Also, explain in the text why the analysis of every palaeocommunity from a 10 million year period is useful for you and include a statement about this in the caption as well. The results themselves aren't surprising or new, so readers need to be told why you have done this.

Line 320-321 – medium sized mammals are the most common large mammals? That doesn't make sense. More to the point, an animal weighing 10,935 kg is definitely not medium sized. How are distinguishing between mediums and large mammals?

Figure 4 – why is every 20th entry selected for visualisation? I don't understand why this was done - it is not a standard way to describe a plot of CA results. This should either be justified or modified.

Line 345 – tiny is not an appropriate word here.

Line 380 – this section needs to acknowledge that no changes to community structure were detected with the very coarse trait categories studied. The scale is relevant.

I hope my comments on the revised draft are useful to you and, as always, I am happy for you to ask for help if any of my points are unclear!

Kris Kovarovic
Durham, UK

(Remarks on code availability)

Reviewer #1 (Remarks to the Author):

Review of Eurasian-African Large Mammal Biogeography: Taxonomic interchange, functional stability, and Late Pleistocene restructuring by Sun et al.

The authors seek to test two hypotheses, namely that Homo was a part of a greater faunal exchange between Africa and Eurasia, and whether the arrival of hominins dramatically changed faunal communities. The authors test these hypotheses by examining community structure defined taxonomically and functionally. These are interesting hypotheses, which have the potential of providing insight into some major events in the evolution of our own species as well as the structure of mammal communities we find today. However, there are a number of points the authors will need to address to adequately test these hypotheses. Some may be surmountable, but others may not be achievable based on the quality of the data available. My major concerns are as follows:

1. The age of each site was calculated as the midpoint between the maximum and the minimum age reported. While this is a common way of dealing with chronological uncertainties for these large modelling efforts, it assumes a normal distribution of age probabilities, which is rarely warranted but mitigated somewhat by using 1 Ma time bins. What's more difficult to accept is that this logic is extended to geological intervals. In many sites examined by the authors, the estimated age of the site is significantly larger than the age bins used in subsequent analyses – at worst, the age range of, for example, Dadingshan, is between 5.333 and 0.0117 Ma, with the midpoint at ~2.7 Ma. There's no reason why this site belongs better in the 2-3 Ma bin than the 5-6, 4-5, 3-4, 1-2, or 0-1 Ma bins. The estimated age needs to be smaller than the bins examined. This means increasing the age bins, eliminating sites, or a combination of both.

Many thanks for the suggestions. We searched for the latest studies on all the sites with a duration of more than 2 Ma in our study and were able to reduce the age uncertainty for many of these accordingly. Additionally, we revised the visualized data such that sites with an age uncertainty of greater than 2 Ma were excluded from appearing in the 1 Ma interval plot. We also excluded sites with age that span more than one epoch period to avoid the mixed genera assemblages. We have now also added a supplementary figure (Supplementary Fig. 3) presenting the data by epoch (Late Miocene, Pliocene, Pleistocene).

2. How the functional groups were allocated and the taxonomic scale at which they were estimated is not, as the authors suggest, “relatively straightforward” (line 299). It's difficult enough at the species level for extant taxa, let alone at the genus level for fossil taxa. Assigning traits in a categorical fashion is probably the only way this can be meaningfully operationalised. However, these traits are not discrete but instead are a continuum, which becomes harder to justify at the genus level across three trophic guilds, and even more so for five locomotory categories for fossil species. For example, Ursus is classified as an omnivore. How is this defined and justified? In modern species, Ursus arctos has an omnivorous diet, but one that is dominated (>90%) by plants (e.g., Ogorstov, S.S. The Diet of the Brown Bear (Ursus arctos) in the Central Forest Nature Reserve (West-European Russia), Based on Scat Analysis Data. Biol Bull Russ Acad Sci 45, 1039–1054 (2018). <https://doi.org/10.1134/S1062359018090145>). In contrast, Ursus maritimus is almost exclusively a carnivore. The authors could argue that the average diet of these two species means the genus as a whole is omnivorous. But this obscures some quite important ecological distinctions. This is exacerbated for locomotion – what distinguishes, for example, a scansorial versus arboreal species? Is the former one that spends a certain percentage of

time on the ground? How then is that calculated for fossil species, some of which may only be known from cranio-dental fragments? At the very least, the authors need to provide detailed justifications for how these were assigned for each genus AND species, and where uncertainties (which will be many by virtue of the nature of the fossil record!) exist. These uncertainties, together with the loss of resolution resulting from examining the traits at the genus level, likely explain some of the strange results the authors get. For example, in the modern communities (Fig 1B), it appears that Bornean mammal communities cluster more closely with Arctic communities than they do with other Indonesian regions! This makes no sense to me and makes me question their fossil results (which as mentioned, will be based on worse data than modern taxa).

Many thanks for the suggestion. The study is based on genus level in order to avoid the uncertainty in species identification. The definition of the categories of locomotion and diet follows the published database and literatures. All the trait data is mostly based on the Paleobiology Database, PanTHERIA, CarniFOSS, Kissling et al. (2014), and Faith et al. (2019). In order to address the reviewer's concern, we revised the diet trait data and added multiple states for the genus when its species showed different states. We assign all the possible categories of the different traits to the genus. For example, now *Ursus* has diet of Carnivore, Omnivore and Herbivore. We then ran random selection for the diet before applying the pairwise dissimilarity analysis. We ran this several times to make sure the randomization of the diet did not have an influence on the result. We also ran the functional clustering analysis based on 2 traits (body mass, diet) and showed the results in Supplementary Fig. 2

3. How do the authors define the continental landmasses? Africa is usually self-evident, but where do the authors draw the line between Europe and Asia? Is that even a meaningful distinction to make? They are both part of the same biological realm, meaning their species share a common phylogenetic history (Holt, B. G. et al. An update of Wallace's Zoogeographic regions of the world. *Science* 339, 74–78 (2013)). Perhaps a clearer definition of these, and what is meant by an "African fauna" for example, would clear up some confusion I had reading through their results and discussion. It was not clear, at least to me, if when using the term African fauna the authors were referring to fauna from sites that are physically situated in that continental region, or sites that clustered most closely with other sites that were predominately in Africa. Put another way, how would the authors refer to a faunal assemblage from a site situated in Eurasia but dominated by African fauna?

Many thanks for the suggestion. We revised the Results and Discussion sections and used the cluster numbers to avoid confusion.

Other minor points to consider:

1. Line 149 – "literature describing individual taxa" – please provide this list

We added the literature references in the supplementary spreadsheet file "trait.csv".

2. It would be useful for the authors to include some sensitivity tests, e.g. using different time bins, more limited functional groups (e.g. one trait, two traits), etc.

We added plots using epochs as intervals (Late Miocene, Pliocene, Pleistocene) and also functional analyses with only two traits (body mass, diet) as supplementary materials. These new visualization and sensitivity analyses show that our main findings are not affected.

3. Figure 1 – I found this figure very hard to interpret. What are the genus clusters 1, 2, and 3? Likewise functional clusters? Also the colour scheme made it difficult to discern some of the patterns.

We changed the title of the legend from “Genus taxonomic cluster/Genus functional cluster” to “Fossil taxonomic cluster/Extant taxonomic cluster/Fossil functional cluster/Extant functional cluster” as well as the color from purple to yellow for fossil data in the plot. We also changed the color for the data for extant mammals in order to more clearly show that the extant and fossil datasets were analyzed separately. We also revised the figure caption to more clearly explain what the different clusters represent.

4. I think the chronofauna analysis needs to be extended. What is the mirror of this analysis doing (i.e. African site as anchor). Does selecting a site with known hominin influence on the fauna change these results (critical I think for really testing hypothesis 1).

The chronofauna is a supplement to the 1 myr temporal bin map visualizations. It provides an indication of the similarity to a particular site's fauna across time. The Nihewan Basin size of SSMZ was previously chosen to visualize whether elements of its faunal development might have experienced major changes around 2 Ma. We have now also added another three sites to this analysis representing the three taxonomic clusters, chosen based on the clustering method PAM (partitioning around medoids). This further illustrates that there was a major faunal turnover in Eurasia while the fauna in Africa rather remained distinct.

5. Line 220 – you have gone to considerable effort to quantify everything, which will allow you to test whether these differences are statistically different, rather than just relying on the patterns in the figures.

NMDS (non-metric multidimensional scaling) is a method used to visualize the similarity among the data points. The patterns shown in the figures are important and meaningful in this analysis. The stress value is the statistical value to check if the result is good or not. We showed the stress value of the nMDS analysis to show how well the ordination fits the relationship of the data points.

6. Figure 4 – This figure could be clarified, consider colour choice and font sizes

We changed the color according to the color change of figure 1 and made the font size bigger to be easy to read.

7. Line 241 – to which figure or results are you referring to here?

It referred to the extant clusters in Fig. 1. We revised and added more texts for a better understanding of figure 1.

8. Line 291 and 324-5 – would be worth reconsidering in light of my comments above. I would argue that, unless shown explicitly otherwise, this is the result of the nature of the fossil record and how the functional groups have been assigned.

As mentioned above, we revised the functional trait scores and ran the analysis several times using randomization We find that our results are overall robust to uncertainties of trait scoring.

9. Line 320 – you have the dataset in place to see if *Homo* is the only genus moving from Africa to Eurasia at this time. Is that the case?

Homo is not the only genus dispersing from Africa to Eurasia during this time. But what we show is that the amount of interchange around this time was not significant enough to be noticed at the continental scale, which led to our conclusion that *Homo* was not part of a larger faunal event.

Julien Louys

We thank you so much for your helpful comments.

Reviewer #2 (Remarks to the Author):

Response to the Authors

This article examines the evolution of taxonomic and functional structure in fossil and present-day large mammal communities from Africa, Europe, and Asia over the past 10 million years. Its objective is to assess whether the dispersal of hominins out of Africa approximately 2 million years ago was part of a broader wave of faunal migration into Eurasia and whether it was accompanied by significant changes in the functional structure of large mammal communities in Eurasia.

The study's results do not support the hypothesis that hominin dispersal was linked to a massive faunal migration from Africa. Instead, they indicate that the most relevant faunal exchanges during the Plio-Pleistocene occurred between Europe and Asia, while African faunas have remained taxonomically distinct from Eurasian ones for approximately 5 million years. Likewise, no evidence was found that the arrival of hominins in Eurasia was associated with significant modifications in the functional structure of large mammal communities. Although the study presents an interesting and potentially relevant approach to understanding hominin dispersal, I believe there are methodological and interpretative issues that should be addressed in greater depth.

Title Recommendation

It would be advisable for the title to more explicitly refer to the ultimate objective of the study, which is to evaluate whether the dispersal of hominins out of Africa approximately 2 million years ago was part of a broader faunal migration and whether it altered the functional structure of large mammal communities in Eurasia.

Thank you for your suggestion. We modified the title to: **Old world mammal paleobiogeography: no evidence for restructuring coinciding with hominin dispersal. We think this better reflects the objectives and findings of the study.**

Materials and Methods

The study aims to detect changes in the taxonomic composition and functional structure of

mammal communities over time. To achieve this, it analyzes the taxonomic and functional similarity of both fossil and present-day communities across time and space. Although previous studies have addressed specific aspects of the evolution and distribution of certain mammal groups in these regions, the combination of taxonomic and functional analyses within a broad temporal and spatial framework represents a valuable and complementary contribution to the existing literature.

Choice of Functional Analysis Method

The functional analysis is based on transforming a dissimilarity matrix using PCoA before applying k-means. However, since the main objective is to identify clear temporal transitions in functional structure, it would be more intuitive to use the number of species in each functional category (size classes, diet, and locomotion) as input variables. This approach would avoid the need to transform the data into an artificial Euclidean space, as functional categories naturally define a Euclidean space. Additionally, it would allow for a more direct identification of transitions in functional composition, facilitating the analysis of changes across key periods.

If the goal were to study more subtle relationships between communities or work with more complex taxonomic or functional distances, the use of dissimilarity matrices and PCoA would make sense, as they capture more nuanced differences between communities. However, if the intention is to detect clear transitions in functional or taxonomic structure over time, the method based on the number of species per functional category appears more efficient and straightforward.

If the current approach was chosen for specific reasons, it would be advisable to explicitly state in the manuscript why this option is preferable to the direct use of functional categories.

We changed the methods accordingly. We conducted the analysis based on the number of genera in each functional category as suggested. We also used beta.pair.abund function in R to conduct the distance matrix on the new functional abundance data we had. The result showed similar pattern as before and did not have strong influence on our conclusions.

Justification for the Choice of k=3 in Clustering

In the manuscript, the selection of k=3 is justified as the minimum number of groups necessary to distinguish the three continents. However, this choice does not appear to be supported by a statistical evaluation of the optimal data structure. Such a low number of groups means that each cluster encompasses significant functional and taxonomic diversity, potentially obscuring important ecological transitions.

Unlike, for example, the Great American Biotic Interchange (GABI), where the formation of the Isthmus of Panama created a land connection that structured dispersal patterns between North and South America, the dispersal of hominins out of Africa cannot be explained by the mere disappearance of a well-defined biogeographic barrier. In this case, if a reorganization of mammal communities occurred, it must have been the result of ecological processes. As a result, changes in taxonomic composition and functional structure of communities should not necessarily manifest at a continental scale but rather in smaller ecological units.

Given that the Euclidean space defined by the approximately 12,000 analyzed communities is extremely large, such a low number of clusters may have prevented the detection of significant transitions in community structure across different time periods. To assess whether the optimal number of clusters has been correctly determined, it would be

advisable to explore standard statistical methods such as the elbow method or the silhouette coefficient, which would allow verifying whether $k=3$ is indeed the most suitable structure or whether a higher number of groups would better reflect the variability in community composition. If the data indicated, for example, that $k=4$ is the optimal number of clusters, forcing $k=3$ could lead to the merging of two distinct groups, making it harder to detect transitions between them.

Additionally, to assess whether there are discrete functional structures within the data, as expected according to Mendoza and Araujo (2022) and González-Trujillo et al. (2024), the application of AMD analysis could be considered (see Supporting information, Mendoza and Araujo, 2022).

We added the AMD analysis on both the taxonomic and functional analysis of fossil and extant data in the study. The result is shown in Supplementary Fig. 1. The result of the fossil taxonomic cluster is 2, while the results of the fossil functional cluster, the extant taxonomic cluster and the extant functional cluster are all 3. The result of the AMD analysis indicated that there are discrete structures in our data.

If there is a solid justification for the choice of $k=3$, it would be advisable to explain it in greater detail in the methodological section, substantiating its validity against other alternatives that might offer a more precise resolution and better capture the dynamics of change in mammal communities.

We have also added a silhouette coefficient analysis in the study to support the determination of the number of k . The results of the fossil and extant taxonomic clusters are 3, while the results of the fossil and extant functional clusters are 2. We used $k = 3$ in our study to show the pattern of the data.

Interpretation of Statistical Analyses

The article employs advanced statistical analysis methodologies, which make it valuable for paleobiogeography. However, it would be beneficial to include a brief, accessible explanation of the function of each analysis and the interpretation of its results. Although the article is intended for a specialized audience, a more intuitive description of each methodological approach would facilitate understanding for readers unfamiliar with these techniques.

We added more descriptions in the Methods section for a better explanation of the methods we introduced in the study.

Taxonomic and Functional Dissimilarity: Missing Key Information

The manuscript states that total dissimilarity consists of two distinct components: replacement and nestedness. However, it does not explain how these components were incorporated into the analyses or how they are interpreted in the results. Since these two components capture different dimensions of dissimilarity between communities, their treatment in the study may significantly influence the interpretation of the observed patterns.

It would be helpful for the authors to clarify whether both components were analyzed separately or if only total dissimilarity was considered. Additionally, it would be relevant to specify whether they play different roles in identifying taxonomic and functional clusters, as

well as whether their relative weight has specific biogeographic implications in different temporal or spatial contexts.

The result in the analysis gives three components, turnover, nestedness and total dissimilarity. Total dissimilarity accounts for both turnover and nestedness together without any weight. It would be better to use total dissimilarity so that no dissimilarity is lost in the study. We have now made this clear in methods.

Furthermore, the manuscript states that the dissimilarity matrix is calculated within each 1-million-year interval and that fossil data are visualized in these intervals. Since the k-means analysis is performed with $k=3$, can the authors confirm whether clustering is applied independently within each time interval, or whether clusters are tracked across intervals? This would imply that cluster structure could change between periods, as community composition varies over time, and that in each interval, communities are assigned to the three clusters solely based on the dissimilarity structure at that moment.

If the goal is to assess the persistence of clusters over time, have the authors considered analyzing to what extent the groups formed in one interval remain in the following ones? Currently, the manuscript does not explicitly mention whether clusters are compared between periods or if they are simply analyzed independently in each interval.

A clarification on this point would be appreciated to better understand the methodology used.

The fossil data and extant data were analyzed separately. The fossil data was not analyzed within 1 Ma interval but rather all together, and only the result was shown in 1 Ma intervals. Therefore, the cluster analysis already includes both temporal and geographic components. We additionally added a new plot in the supplementary file visualizing the data in epochs (Late Miocene, Pliocene and Pleistocene) (Supplementary Fig. 3).

Structure of the Materials and Methods Section

Currently, the structure of the Materials and Methods section does not clearly reflect the relationship between the four methodological approaches used in the study (taxonomic clustering, functional analysis, chronofaunas, and nMDS). The inclusion of specific subsections for each of these approaches, organized coherently and with direct references to the figures presenting their results, would significantly improve the clarity of the study design.

We changed the Methods and Results sections in the study, accordingly, adding subsection headers for each of these approaches.

Greater Clarity in the Description of Analyses

Providing more details on how each method is applied and its specific role in the analysis would help improve clarity and facilitate the interpretation of results. In particular, it would be useful to explain why each method is employed, what specific problem it seeks to address, and what type of information is obtained at each step of the procedure. This would better contextualize the results within the study's objectives and provide a more precise interpretation of the observed patterns.

We revised the Methods section in order to explain how each method works and the type of information it provides.

Results

Structure of the Section

As in the Materials and Methods section, the structure of the Results section does not clearly reflect the relationship between the four methodological approaches used in the study. Including specific subsections for each of them, organized coherently, would significantly enhance the clarity of this section.

We changed the Results section and added subsection headers to make it clearer. Now the Result section has four subsections: fossil taxonomic analysis, fossil functional analysis, extant taxonomic analysis and extant functional analysis. The fossil analyses contain results of pair-wise clustering, nMDS and correspondence analysis in order. The extant analyses contain pair-wise clustering.

Greater Depth in the Analysis and Interpretation of Results

The description of the results is often too brief, making it difficult to follow and understand. It would be advisable to provide more detail on the findings, ensuring that the presentation of the data is clear and complete before their interpretation in the Discussion section.

We revised the Result section to provide more details.

Clarity in Figure Captions

The figure captions in the manuscript are too general and do not adequately differentiate the various panels within each figure. This hinders the interpretation of the results. It is recommended to expand the descriptions to include more detailed and specific information about each panel.

We revised the figure captions and legends; they now include more information and are more detailed.

Interpretation of Results and Consistency with Figures

In the Results section, some statements do not seem to be fully supported by the data displayed in the figures. Below are specific inconsistencies along with suggested reformulations:

- Line 191. The statement: "The three taxonomic clusters generally corresponded to a Miocene African-Eurasian fauna, a Miocene-Pleistocene African fauna, and a Pliocene-Pleistocene Eurasian fauna (Fig. 1)" does not seem to reflect the spatial patterns observed in the maps. A more precise reformulation could be:

"The three taxonomic clusters generally corresponded to a predominantly Eurasian fauna in the Late Miocene (Cluster 1), which is now found in North Africa, the Middle East, and South Asia; a fauna that was present on all three continents until the Early Pliocene, when it became restricted to Africa, and is currently found in sub-Saharan Africa (Cluster 2); and a fauna that emerged in Europe in the Late Miocene, gradually expanded into Asia, and currently dominates Europe, northern Asia, and parts of eastern Asia (Fig. 1A)."

We changed the color in Fig. 1 in order to avoid misunderstanding. And we changed the text and the order in the Result section in the paper accordingly, now the fossil data and extant data are in different subsections.

- Line 193. The statement: "From 10 Ma to 6 Ma, the African and Eurasian sites shared a

very similar faunal composition..." does not appear to be supported by Figure 1A, where the Eurasian cluster (Cluster 1) is present in Europe and Asia but is almost absent in Africa.

The reviewer is correct. There are very few African sites between 10 and 6 Ma, but still the majority of these belong to a different cluster than the Eurasian sites. We have changed the text accordingly to indicate that the African fauna and Eurasian fauna is not similar from 8-9 mya.

- Line 197. The statement: "This second cluster became dominant in Africa from 5 Ma until today, distinguishing African faunas from those in Eurasia since that time." does not align with the distribution observed in Figure 1A. To claim that a cluster became dominant, one would expect a more widespread distribution across the continent, which does not seem to be reflected in the data shown. A more precise reformulation could be: "This second cluster is present in all available sites in Africa from 5 Ma until today, distinguishing African faunas from those in Eurasia since that time."

We rewrote the Discussion section and changed the text accordingly.

- Line 202. It is mentioned that between 6–5 Ma and 4–3 Ma, Asian faunas still belonged to the "Miocene" cluster (Cluster 1). However, during this period, there are few Asian localities with sufficient data, and these exhibit all three cluster types. This suggests that the statement should be nuanced to better reflect the observed distribution.

We changed the text to "Cluster 2 started to appear during 6 to 5 Ma and gradually occupied Eurasia later with cluster one finally disappearing around 3 Mya."

- Line 253. The claim that Eurasian and African faunas showed high similarity between 10 and 7 Ma is not consistent with what is observed in the maps. In Figure 1A, the Eurasian cluster (Cluster 1) is well represented in Europe and Asia but has little presence in Africa.

We have changed the text accordingly to indicate that the African fauna and Eurasian fauna is not similar from 8-9 mya.

- Line 255. The idea that a distinctive African fauna emerged between 7 and 6 Ma should be nuanced, as Cluster 2 was already present in earlier periods. Rather than an abrupt change, it seems to be a gradual process in which the differences between continents became more pronounced over time.

We changed the text to: Though data for this time interval is sparse, African faunas appear to be distinct from Eurasian faunas, by at least 8 Ma.

- Line 256. It is mentioned that between 6 and 5 Ma, a new cluster appeared in northwestern Europe (Cluster 3), which replaced more archaic Asian faunas during the Pliocene. However, the maps show that the previous clusters continued to coexist for some time in different regions, suggesting that the transition was more gradual than the text implies.

Cluster 1 existed in Eurasia for a long time while cluster 2 appeared during 6 and 5 Ma then gradually occupied Eurasia later. But in general cluster 2 gradually occupied Eurasia in the Pliocene and almost dominated Eurasia in the Pleistocene. We changed the text precisely in the paper according to the new result. The revised text now reads: "Between

6 and 5 Ma, a new fauna appeared in northwestern Europe that replaced more archaic Asian faunas during the Pliocene, and from which modern Eurasian faunas are derived.”

It is recommended to reformulate this section to more accurately reflect the spatial distribution of clusters in each period and avoid overly simplistic interpretations of faunal composition changes.

We have revised the Methods and Results sections in the paper to make the connection of Methods and Results section clearer and provide more detailed Result section.

Conclusion

Overall, this study addresses a relevant question about the relationship between hominin dispersal and the dynamics of mammal communities over time. However, several methodological and interpretative aspects require further development and justification. The selection of the number of clusters without clear statistical support, the brief methodological explanations, and the lack of correspondence between some results and the figures affect the robustness of the conclusions.

To improve the clarity and rigor of the study, it would be advisable to include more detailed justifications for the analytical decisions made, restructure the presentation of results to more accurately reflect the observed patterns, and expand the discussion on the biogeographical and ecological implications of the findings. Addressing these aspects would significantly strengthen the contribution of the article and facilitate its interpretation within the broader framework of mammal community evolution and hominin dispersal.

I hope my feedback contributes to strengthening the manuscript.

Best regards,

Manuel Mendoza

We thank you very much for your helpful comments.

Reviewer #3 (Remarks to the Author):

I have read the study by Sun et al. with great interest as the topic of hominin co-migration with other fauna and their potential influence on the functional composition of large mammal communities is of central importance to understand the onset and extent of hominin impact on global ecosystems. As I am not an expert in the methodology used in the study, I cannot provide fair judgment on the quality of the applied methods. However, testing of the two main hypotheses outlined in the paper (hominin-mammal co-migration and hominin impact on mammalian functional composition) is mostly studied by examining fossil evidence from non-hominin mammals and it thus seems that the proposed hypotheses are tested tangentially at best. Given the focus of the proposed hypotheses on hominin-related dynamics, a more direct analysis of the relationship between hominin fossils and other mammalian fossil assemblages is needed. For example, as some fossil assemblages contain fossils from the Homo genus, it would be instructive to test if this Homo presence in any way affected the taxonomical and functional composition of the rest of the assemblage (compared to assemblages that do not contain Homo remains). While the global trends indicate that large taxonomical and functional changes occurred either before 2 mya or during Late Pleistocene, this does not exclude the possibility of hominin

impact at a more local scale (testable by comparing Homo-containing and non-containing assemblages, as mentioned above). Additionally, to provide a stronger case for Homo not affecting mammalian biogeography, climate or similar data that have been reconstructed for the studied period could have been used to test if environmental shifts drove the observed patterns.

Homo was included in the data in our analysis. In order to improve visualization, we now marked all sites that include *Homo* in Fig 1. Ultimately, however, our data is used to examine large-scale changes at the continental scale, and do not exclude the possibility of hominin impacts on taxonomic and functional turnover at more local scales. We agree that it would be helpful to include environmental data in future studies.

In general, I would like more elaboration on the selected methodology and data selection. Specifically, I agree that body mass and diet likely capture the most fundamental functional roles of taxa in a community, but I am unsure how locomotion would be informative (especially as it is defined with respect to habitat type, which may or may not reflect locomotion as the ability to move from one place to another). Additionally, all life history traits are correlated and I am uncertain whether the methodology used adequately accounts for the strong correlations among these traits and their potential impact on the observed patterns. Generally, the methodology needs further justification to ensure it is more self-contained and the choices made are clearly explained. More specific details follow below.

Following suggestions by other reviewers as well, we have provided more information in the Methods section, and also conducted new functional pairwise dissimilarity analyses, as well as sensitivity tests now shown as supplementary material.

Line-by-line comments

Line 103: Cantalapidra should be Cantalapiedra

Corrected

Line 134: Current IUCN ranges are heavily influenced by human impact - therefore it should be addressed how this may impact the analysis. Additionally, mammalian present natural ranges from the PHYLACINE database (10.1002/ecy.2443) may be more appropriate to use.

We used IUCN data according to the method from Holt et al., 2013 where they used data from IUCN for zoogeographic regions. However, we have now added analyses using both 'current' and 'present natural' datasets from the PHYLACINE database. The results are broadly similar to those using IUCN data and are shown as Supplementary Fig. 4.

Line 153: More detail is needed about how the "pairwise distance matrix" is calculated, rather than a simple reference to an R package.

We added more details in the Methods section

Line 158: "k=3" needs to be justified by a plot of variance explained vs. the number of clusters, a simple "we found" statement is insufficient

As noted above, we added a silhouette coefficient analysis to examine the best number of k, which showed that for the fossil taxonomic clusters, the best number is 3, while for the fossil functional clusters, the best number is 2. We decided to use k=3 which suits the result of the silhouette coefficient analysis for the fossil taxonomic clustering.

Lines 161-168: A single reference site is chosen as a point of reference. However, I suggest multiple similar points should be tested to show robustness of the observed patterns.

We added three more sites to the chrono-faunal analysis, chosen based on the median of each cluster.

Lines 169-170: The “genus-trait matrix” and the “Gower dissimilarity matrix” need to be defined.

We changed the method for the functional analysis according to Reviewer 2’s advice to now use the number of genera in each functional category as suggested. We then used `beta.pair.abund` function in R to conduct the distance matrix on the new functional abundance data we had. The result showed similar pattern as before and did not have strong influence on our conclusions.

Lines 175-176: Why was the same number of clusters used in functional analysis as in the taxonomic analysis? The functional traits contain many categories and thus many more combinations of traits are possible than 3. I therefore do not see how “following the same principal of the taxonomic clustering” is justified.

The newly added silhouette coefficient analysis indicates that the best number of k for the fossil taxonomic clusters is 3, the fossil functional clusters is 2, the extant taxonomic clusters is 5 and the extant functional clusters is 2. To permit easy comparison, we set all of these four analyses to k=3. We now provide the results of the silhouette coefficient analysis in the supplementary text.

Line 200-201: The sentence “Homo was present in 22 out of 162 sites.” seems awkward in this position as it is not followed up with a reason why this might be an important statistic to report.

We deleted this sentence.

Line 213: “difference peaking”? - should it not be “similarity peaking”? Additionally, the peak is only clear for Asia, after which it drops off towards the Late Pleistocene (an explanation for this trend might be good to include), while the similarity reaches a plateau for Europe in the Mid to Late Pleistocene. Additionally, an explanation of the lines and shaded area would be needed to better understand Figure 2.

“Difference peaking” indicated that both Asian and European curves reached highest difference comparing to African curve in our data in Pleistocene. The whole sentence “In the Pliocene, similarity curves diverged as European and Asian faunas developed greater similarity to SSMZ than African faunas, with the difference peaking in the Early Pleistocene.” is comparing the curves between Eurasia and Africa, so “difference peaking” should be correct here. We added more description to this section: “In the Pliocene, similarity curves diverged as European and Asian faunas developed greater similarity to SSMZ than African faunas, with the difference peaking in the Early Pleistocene. Persistently low similarity of the African fauna with SSMZ contrasts with the European and Asian faunas through this time, suggesting little to no dispersal of genera from Africa to Asia. The peak in Asia indicated that the highest similarity occurred at the beginning of Pleistocene and started to go down in Pleistocene, while in Europe there were more similar sites emerged. We also added three other sites here. We added more captions for this figure and explained the gray area.

Line 220: The separation between geographic clusters (Fig. 3A) is described as strong. I think this needs to be attenuated as clustering is not very clear, especially during the Miocene.

We rewrote this section to more clearly explain that African and Eurasian sites are clearly separated comparing to European and Asian ones.

Line 241-246: The paragraph argues that extant taxonomic and functional clustering “followed a similar latitudinal pattern”, which is not very accurate given the maps in Fig. 1A,B.

We changed this part of the text to: “The functional clustering also showed a latitudinal pattern”

We show our many thanks to reviewer #3

Reviewer #4 (Remarks to the Author):

Thank you so much for asking me to review this manuscript, which I approached with great interest and enthusiasm! I am happy to disclose my identity to the authors, and if there are any questions about my comments in this review, I would welcome direct communication.

This study is generally great, and I would like to see it published in some form. The data collection alone must have been a feat; despite the increasing availability of fossil data through public databases such as NOW and the PBD, these data are not easy to merge and process. Additionally, I note that new Asian site data is included herein – 92 new previously unpublished sites, if I have understood it...? It's not clear how these data were collected or acquired, but I would like to suggest that this individual(s) is included in the author list. The compilation of new data is a major contribution to science and if the data hasn't previously been published elsewhere then I recommend that the data contributor join the author list.

All the data newly compiled for the study were collected by the lead author from the literature. The raw data itself has been made available as a supplementary file “fossil.csv”, and will form part of the BICAEHGIS database, which will be made public shortly.

The major comment I wish to start with is that this manuscript contains a large-scale study of major shifts (or not, as the case may be) in the taxonomic composition and functional structure of mammalian communities over ten million years, in one-million-year intervals. The time scale of this may be at odds with the scale required of a study of hominin migrations out of Africa. This is not a fatal flaw in my opinion! But, somehow this paper needs to be framed in a different way. The introduction details two major phases of hominin dispersal out of Africa, emphasizing the first one around 2 mya. The hypotheses are both testing issues relating to these dispersals. This gives the impression that the study will focus on the time periods relevant to these dispersals and although they are incorporated into the study, they are not the analytical focus. Therefore, my advice would be to reframe – the intro should be more ‘honest’ about the ensuing analysis, centering mammal communities and their changes over long period of time, rather than hominin dispersals from 2 million years onwards. But I appreciate that hominins may be thought of as a greater point of

interest by some – not me, as in my opinion a study of mammal communities over 10 million years (during which hominins happened to disperse out of Africa) is in and of itself an excellent study! But if the authors did wish to truly make the focus of the entire paper about hominin dispersals and their (potential) correlation with changes in mammal communities, I would say that the earliest time periods could be removed from the study as they aren't pertinent, and the analyses should look at a much finer grained time scale to better capture the time when hominins disperse and arrive in new places (and indeed enough time goes by to see any changes that might occur in the communities as a result). In my opinion this would take far more work than reframing the study to focus on mammal communities over time, painting their structure and function with a broad brush to look at macro patterns. Within that framework the authors can comment on events during that long timeframe that have the potential to modify communities including, but not limited to, hominin dispersals.

Many thanks for the advice. In this study, we aim to explore the driving factors and influences on the large mammal communities and hominins out of Africa. You are correct that a finer-scale study (in terms of temporal and maybe also geographic coverage) could have addressed the question at higher resolution. However, we intentionally designed our study at the coarse scale here (our choice of genus level, and the three traits also reflects this). The time scale is set to 10 Ma so that we can have a better and larger view of the whole large mammal community over Eurasia and Africa. As you point out, the objective need not have focused on hominins, but in this case this does in fact honestly reflect the objective of the study from the beginning (part of the larger BICAEHFID Project). Reframing the intro part would therefore result in a loss of focus on hominins dispersals, which is an important part of this study.

Below I outline additional comments and questions associated with relevant line numbers. Since I have suggested reworking the framework and hypotheses, it is difficult to direct some of my comments at a hypothetical newly framed paper. Therefore, it is probably easier if I provide my comments in the order they came to me whilst reading the manuscript. I have five substantive questions about the methods because I wasn't clear on what all of the analyses accomplished in terms of testing the two stated hypotheses. I don't make many comments about the discussion section because the paper's reframing would necessitate a new discussion, and the thoughts that came to mind were more or less the same as those that arose when I read the introduction (about the timescale and granularity of the study), and which are described above.

Line 33 or thereabouts – you could acknowledge the newly reported 1.95 Mya Romanian material by Curran et al 2025 (<https://doi.org/10.1038/s41467-025-56154-9>)

We added it in the texts and also revised the faunal data in our dataset.

Lines 47-49 – this paragraph is only a single sentence describing the 2nd OoA event, which isn't consistent with the summary of the 1st event. Could more be added for better context?

Here we concluded the driving factors of the dispersal of *Homo sapiens*. To keep it clear and concise, we would like to keep it this way.

Lines 50-63 – Could this paragraph be rewritten so that it reads more smoothly? I also recommend that you delete the word natural (palaeoenvironments are of course natural, so

that word is redundant) and delete the word huge (which is a bit colloquial).

We rearranged the sentences to make it smoother and deleted the words as suggested.

Lines 64-83 – the first part of this paragraph refers to herbivores and then carnivores, so I suggest you stick with this order throughout the paragraph and review the herbivore info first, then move onto carnivores. You could then more easily link to the next paragraph (lines 84-88), which stands a bit awkwardly on its own (or better yet, incorporate it).

Here we didn't quite understand the reviewer's suggestion. The order is first all mammals, then herbivores, then carnivores, followed in the next paragraph by further discussion of carnivores.

Lines 99-115 – this paragraph mostly consists of paper-by-paper one sentence summaries, rather than a synthesis. Also, and with the caveat that this is a stylistic point and perhaps the authors don't find fault with it, but nearly each sentence begins with the name of an author(s). The text shouldn't be about the people, it should be about their science. This style may be what has got you stuck in paper-by-paper summaries, as well. I would recommend that this paragraph is rewritten so that it doesn't name other researchers throughout (or that this practice is reduced) and the material is better integrated.

The paragraph has been rewritten.

Lines 116-122 - Could this paragraph be linked more clearly with the above paragraph(s)?

This is the paragraph for introducing the hypotheses used in this study.

Line 134 – geographic distribution data comes from IUCN and although this would work for extant taxa in quite recent history, I'm not sure it works well for taxa in the deeper past, even if they are extant. For example, Thailand has had a number of local extinctions during the Pleistocene, but many of these taxa live elsewhere in Asia today (pandas and orangutan). So, the IUCN data won't provide the correct distribution data for these species. This is my first major query about the methods in this paper. I'd like to know how this issue was accounted for or if it could be (or evidence to support that there is no need to).

As noted in response to a previous reviewer's comment, in our revision we also conducted our analyses on the PHYLACINE dataset, which includes both 'current' and 'present natural' datasets. The latter estimates geographic ranges in the absence of human impacts. Results of the PHYLACINE analyses are similar to those using IUCN data (shown as Supplementary Fig. 4)

Lines 145-147 – are these the same diet and locomotion categories used by others, so you are following an established system?

Yes. The sources of the trait scores are listed in the same paragraph. We also added the reference in the supplementary spreadsheet file "trait.csv"

Line 149-150 – It would be more succinct to simply state what the analysed body categories were rather than explain that small mammals were excluded after their body size categories are given earlier in the paragraph.

For genera that span more than one size category, the exact value is determined by randomly choosing a single value early in the analysis. The exclusion of the body mass categories 1 and 2 therefore needed to be done after the randomized choice of the body mass (all shown in the R script provided).

Lines 158-160 – how did you find that $k=3$ is the minimum number of clusters needed to distinguish the continents? That rather makes intuitive sense since there are three continents, but the evidence to support this statement should be provided. Is it in the supplementary info, but I missed it? This is my second major methods-related question. Also, the supplementary info should be referred to wherever it includes data not summarized/presented in the main text. There was a lot of missing info such as the names of the sites, their constituent fauna, their geological dates, their references, and info about the trait assignments for each genus, and those references. There are files pertinent to these data available to download, but they should be referred to in the text.

We now conduct a silhouette coefficient analysis for determining the value of k . The best value of k for fossil data is 2 and 3 respectively for taxonomic and functional clusters. The best value for extant data is higher. But our aim was to compare the differences between the extant and fossil data, so we determined that all values of k should be 3. We provided more plots in the Supplementary as supplementary material. Additionally, we added the reference for the supplementary material in the main text.

Line 177-178 – Add a statement of what nMDS was used for at the end of the sentence, like “in order to…” which circles back to the stated hypotheses. It generally feels like the methods should all be more explicit in how each analysis ties to the stated hypotheses. It was hard to tell how each test contributed to the whole. This is my third major comment on the methods.

We rewrote the Methods section and added more details and explanations on the method we used in the study. We hope it is clearer now.

Lines 191-207 – figure 1 is referred to, but it has two parts, A and B, which need to be referred to where relevant.

We rewrote this paragraph and added the reference to the figure parts.

Line 208 – Figure 1 itself should have the colour legend defined in it, and also in the caption that goes with it. Also, for the Present, it looks like you included areas of very high latitude where you don't have fossil sites represented. Have I read that correctly? Would it make sense to have a latitudinal cut-off for the Present that matches where the fossil sites are?

It is true that very high latitude sites are missing in the fossil dataset. In the analysis of extant function similarity, the entirety of Cluster 1 is located in these high latitudes, for which there is no comparable data. However, in the analysis of taxonomic similarity, these high latitude areas share a cluster with low-latitude regions in SE Asia. This wide pan-Eurasian similarity is also reflected in the fossil data (even without the high-latitude sampling). We think these differences (between taxonomic and functional) and similarities (between fossil and modern) are interesting and worth showing, and therefore we prefer to keep the extant high-latitude sites. We changed the legend for Fig. 1 and added more detailed captions for it.

Lines 210-216 – I must confess that I am not that familiar with chronofauna analysis and that I struggled to get my head around the purpose of it here. I have to ask if it is necessary to the overall story. In looking at the results in figure 2 it looks like the Asian and European fauna follow the exact same curve and that they remain the same distance apart until just before 2.5 million years ago when they diverge from each other at the start of the Pleistocene. So, they both grow increasingly dissimilar to SSMZ until their curves diverge and Asia starts to look more similar to it. This isn't commented on, but it merits an

explanation, particularly the timing. This is my fourth major comment on the methods.

Good point. The chronofauna visualization is in reference to a single site, and for our broader continental-scale perspective it is not necessarily relevant to describe the differences of the geographic curves in detail. Furthermore, we have now added three further site comparisons. The goal of these visualizations is to provide further support to the distinctiveness of the African assemblages in relation to the Eurasian ones, and to show the closer similarity and turnover of European and Asian assemblages among each other.

Lines 230-238 – it wasn't until these results of the correspondence analyses that I realised I was unclear about how the time variable was considered in them. I don't think it was accounted for, am I right in this? If that is so, everything has been lumped together over a 10-million-year period...? To me, that doesn't make sense. Firstly, it is known that taxonomically these three continents differ from each other, but that Europe and Asia share more similarities. So, this doesn't seem entirely new, but a confirmation of what is known - but it would be insightful if the differences were explored over time to look for more nuanced trends (although I suspect there would be a high level of differentiation throughout the time sequence). Secondly, my feeling is that lumping the trait data together for 10 million years was likely to result in a lack of differentiation between the three continents, and again there would be more nuance and insights if the data were analysed by time bin. This is my fifth major methods comment.

Yes, the fossil data are analyzed together in the correspondence analysis. The reason for this is to provide visual information on which taxa and traits have the strongest influence on (i.e. most strongly define) the three clusters determined by the clustering analysis. Analyses by 1 myr time bin, whether for taxonomic or functional data, are not feasible at this scale of analysis as the data would be far too few. An analysis at this smaller timescale would also require using species-level taxonomy and a more refined set of traits might be able to do this (e.g. Faith et al. 2019 www.pnas.org/cgi/doi/10.1073/pnas.1909284116)

Line 290 – these faunal geographic regions are not denoted in figure 1A where the reader is referred to, and have not been mentioned elsewhere. It's true that some readers will be familiar with them, but some sort of explanatory statement is necessary.

We changed the text to " These are similar to classical zoogeographic regions based on the distributions of amphibians, birds, and mammals, with identifiable Afrotropical, Saharo-Arabian, and a combination of Palearctic, Sino-Japanese Oriental regions (Fig. 1A)." to explain how the regions are defined.

Line 298 – "Data inaccuracy is unlikely..." That is a strong statement! There are surely a host of inaccuracies in the data which have not been acknowledged (which are not the authors' fault, but simply the reality of working with fossil fauna community data)... taphonomy, depositional context, difficulties with genus identification, sampling etc. Aspects of the methods could also confound the analyses – for example, the dietary categories are blunt, and include the category 'omnivore' which is something of a catch-all, and sites with only five unique genera were included, which is a very small number. Other researchers have grappled with this latter issue, Louys et al. 2009 finding that 12 taxa was an acceptable minimum number (<https://doi.org/10.1016/j.jas.2009.06.012>).

Our point was that assigning body mass, diet, and locomotion using our coarse categories was largely uncontroversial for the majority of the taxa. We reworded this part to try make this clearer: There are possibly two explanations for these differences: 1) The fossil datasets may be severely inaccurate or incomplete; 2) A fundamental rearrangement of functional groups appeared in the time between the fossil and modern dataset, i.e. during the Late Pleistocene and Holocene. We think it is unlikely that this pattern is the result of inaccurate data, particularly given that the attribution of such fundamental traits as size, diet, and locomotion to both fossil and extant genera was in mostly cases done with a relatively high degree of confidence.

Also in Louys et al. 2009, their method and aim were quite different from our study. They determined a minimum number of species for distinguishing different habitats - which is not an objective of ours. As a comparison, Kaya et al. 2018 (<https://doi.org/10.1038/s41559-017-0414-1>) also used a minimum of five taxa identified to the genus level for their biogeographic similarity analyses

Finally, the Discussion seems somewhat under-referenced, with only about 8 citations. I appreciate that there is a suggested limit to the number of references in this journal, but wonder what can be done to make sure that the relevant body of literature is adequately acknowledged?

We added 2 more key citations (Faith, J. T., Rowan, J. & Du, A. Early hominins evolved within non-analog ecosystems. *Proc Natl Acad Sci USA* 116, 21478 – 21483 (2019) and Martínez-Navarro, B., Antonio Pérez-Claros, J., Palombo, M. R., Rook, L. & Palmqvist, P. The Olduvai buffalo *Pelorovis* and the origin of *Bos*. *Quat. res.* 68, 220 – 226 (2007)) in the Discussion section to strengthen our context and arguments.

Again, I reiterate my belief that this is a wonderful study! Although some aspects need to be ironed out and I have many questions and suggestions, these are offered as constructive pieces of advice. I hope that they do prove helpful in the revision process, and I am truly looking forward to seeing this work in publication.

Many thanks for your comments, which have been really helpful in improving our paper.

Kris (Fire) Kovarovic
Durham University, UK

Reviewer #4 (Remarks on code availability):

The R code is available but I am not a competent R user so I have not reviewed it.

REVIEWERS' COMMENTS

Reviewer #1 (Remarks to the Author):

My major concerns have been addressed by the authors, who have provided additional data and refined their analyses. No further comments from me.

We would like to show our many thanks for your comments.

Reviewer #2 (Remarks to the Author):

Overall, my concerns have been satisfactorily addressed and the revisions improve the manuscript. Three essential adjustments remain: (1) correct the definition of AMD to Average Membership Degree throughout the manuscript, including the relevant figure caption(s) (e.g., Supplementary Fig. 1) and all supplementary text; and (2) remove any statement that $k = 3$ was chosen “because there are three continents” and replace it with the data-driven justification (silhouette and AMD).

If these minor points are addressed, I am happy to recommend the manuscript for publication in Nature Communications.

We changed the text throughout the manuscript and the supplementary document according to the suggestions.

We appreciate your suggestions and would like to thank you for them.

Reviewer #3 (Remarks to the Author):

The manuscript has improved considerably. However, some minor revisions are still necessary. For example, the manuscript should be structured so that figures are referenced in the correct order (e.g., at present, Fig. 4 is cited after Fig. 1 but before Figs. 2 and 3). In addition, the language requires further refinement. I recommend that the manuscript be thoroughly revised by a native English speaker. Additionally, I recommend providing the original code used to analyze the data and generate the figures, if it has not already been supplied.

Reviewer #3 (Remarks on code availability):

No code detected

Many thanks for the advice. We have improved the language, changed the order of the figure citations to the correct order. The R script used to run all analyses is provided as a supplementary file.

Reviewer #4 (Remarks to the Author):

Thank you for the opportunity to review this manuscript in its revised form. I appreciated being able to also read the other three reviewers' comments, through which it became

evident where we had similar suggestions or points of confusion.

In my second round of comments below, I have copied and pasted my initial points with the authors' replies, and I respond directly to each point here (I didn't do this where I had no further comment - only for issues on which I would like to follow up). Hopefully, this makes it easier to follow the review process as a conversation, and I am more than happy to continue this conversation as necessary. After that, I have added new comments with line numbers referring to the revised manuscript.

I can see where meaningful changes have been made to the methods and the processing/treatment of the data (randomisation was a great idea, there is a sensible approach now to putting taxa in the time bins, amongst other improvements). But, the manuscript itself does not make these improvements clear to a reader. The responses made to the reviewers provide a much better explanation of the changes, and why they were implemented, than the manuscript which is not well-written in places (grammatical mistakes and the use of colloquial language – “tiny” “basically”) and does not synthesise the background information or the results for an impactful discussion. It needs to be heavily edited so that the writing, structure and clarity are all improved. The main text of the manuscript is at the 5000-word limit, and I can certainly appreciate the challenges of packing so many analyses into that limit, but to achieve the quality of manuscript that I would expect in Nature Communications, there is much work to be done.

We have significantly rewritten the paper, improving the language and structure.

My other major comment is that you have decided not to undertake the reframing of the paper as I suggested and, unfortunately, I don't think it is helping you “sell” this study very well. Some of my comments below will attest to this. The broad categorisations of fauna that you have used and the ten-million-year time period covered do not support the aims of your paper as they are currently described. Some of the results are not new or surprising, and the results specifically from the relevant hominin timeframe should be explored with much greater depth. Please consider how to appropriately and effectively introduce the main themes of this paper, and link them clearly to the datasets analysed. There is a disjunct between them at present.

Thanks for the advice. In this study we would like to explore the larger context of hominin dispersals out of Africa, and we carefully formulated two testable hypotheses which we believe are appropriate and relevant to that context. As part of the BICAEHFID-ERC project (Biogeographic and cultural adaptations of early humans during the first intercontinental dispersals), our study was explicitly designed with this question and hypotheses in mind from the beginning. Nonetheless, we have improved the text, including the title of the paper, such that the focus on hominins is highlighted alongside the broader biogeographic story. We would like to thank you for the suggestions which gave us a deeper thought of our study. We replied to the concerns below point by point.

FOLLOW UP ON INITIAL POINTS AND RESPONSES:

This study is generally great, and I would like to see it published in some form. The data collection alone must have been a feat; despite the increasing availability of fossil data through public databases such as NOW and the PBD, these data are not easy to merge and process. Additionally, I note that new Asian site data is included herein – 92 new previously unpublished sites, if I have understood it...? It's not clear how these data were collected or acquired, but I would like to suggest that this individual(s) is included in the author list. The compilation of new data is a major contribution to science and if the data hasn't previously been published elsewhere then I recommend that the data contributor join the author list.

All the data newly compiled for the study were collected by the lead author from the literature. The raw data itself has been made available as a supplementary file "fossil.csv", and will form part of the BICAHEGIS database, which will be made public shortly.

If the 92 Asian sites referred to as 'new' were in the literature and available for compilation by the author, then they are not in fact new. They are published works. This description needs to be changed in the text. They are simply 92 sites that form part of the Asian site dataset. The same goes for the description of the Romanian site that was published after the initial submission and then added to the analyses that form the most recent version of the manuscript.

Thank you for the suggestion. We changed the text to "additional" in the manuscript.

The major comment I wish to start with is that this manuscript contains a large-scale study of major shifts (or not, as the case may be) in the taxonomic composition and functional structure of mammalian communities over ten million years, in one-million-year intervals. The time scale of this may be at odds with the scale required of a study of hominin migrations out of Africa. This is not a fatal flaw in my opinion! But, somehow this paper needs to be framed in a different way. The introduction details two major phases of hominin dispersal out of Africa, emphasizing the first one around 2 mya. The hypotheses are both testing issues relating to these dispersals. This gives the impression that the study will focus on the time periods relevant to these dispersals and although they are incorporated into the study, they are not the analytical focus. Therefore, my advice would be to reframe – the intro should be more 'honest' about the ensuing analysis, centering mammal communities and their changes over long period of time, rather than hominin dispersals from 2 million years onwards. But I appreciate that hominins may be thought of as a greater point of interest by some – not me, as in my opinion a study of mammal communities over 10 million years (during which hominins happened to disperse out of Africa) is in and of itself an excellent study! But if the authors did wish to truly make the focus of the entire paper about hominin dispersals and their (potential) correlation with changes in mammal communities, I would say that the earliest time periods could be removed from the study as they aren't pertinent, and the analyses should look at a much finer grained time scale to better capture the time when hominins disperse and arrive in new places (and indeed enough time goes by to see any changes that might occur in the communities as a result). In my opinion this would take far more work than reframing the study to focus on mammal communities over time, painting their structure and function with a broad brush to look at macro patterns. Within that framework the authors can comment on events during that long

timeframe that have the potential to modify communities including, but not limited to, hominin dispersals.

Many thanks for the advice. In this study, we aim to explore the driving factors and influences on the large mammal communities and hominins out of Africa. You are correct that a finer-scale study (in terms of temporal and maybe also geographic coverage) could have addressed the question at higher resolution. However, we intentionally designed our study at the coarse scale here (our choice of genus level, and the three traits also reflects this). The time scale is set to 10 Ma so that we can have a better and larger view of the whole large mammal community over Eurasia and Africa. As you point out, the objective need not have focused on hominins, but in this case this does in fact honestly reflect the objective of the study from the beginning (part of the larger BICAEHFID Project). Reframing the intro part would therefore result in a loss of focus on hominins dispersals, which is an important part of this study.

Most readers will not know what the larger BICAEHFID project is and that it focuses on Homo dispersals (I had to Google it myself), so that rationale will not support the current structure or narrative of the manuscript. If your intent was to focus entirely on human dispersals, there were better ways to analyse the data that would have allowed you to identify more nuanced differences in the communities over the shorter timeframe that applies to these dispersals rather than a 10-million-year view. This was recognized by other reviewers; the broad brush of the dietary categorisations is one such example, as is the use of genera rather than species. There are very good reasons for your choices, but they are particularly good choices for studying a long time period rather than a shorter timeframe like the one that relates to hominin dispersal.

Thank you for the advice. The study aim was to use large mammal data to analyze the overall context of hominin dispersal, thereby shedding light on any possible driving factors. The reviewer is correct that more finely resolved temporal and taxonomic scales provide far more insight. The problem, however, is that the temporal and taxonomic resolution of the Eurasian and African fossil record are not of sufficient quality for this. Historically different taxonomic approaches and practitioners in Europe, east Asia, and Africa means these records are difficult to compare at the species-level without further work to assess possible species synonymies. Similarly, the age resolution of most Eurasian (and many African) Miocene and Pliocene sites does not allow for finer than 1 million-year or 500 kya time bins at best. Our choice of traits (including broad dietary characterizations) is a conservative approach given the quality of the data. Note that our functional analyses do recover strong geographic partitioning in the modern community data, indicating that our combination of trait categories is indicative of actual community ecological structure (and correlated to climatic patterns). We believe the questions we are asking are appropriate for the quality of the data.

Line 33 or thereabouts – you could acknowledge the newly reported 1.95 Mya Romanian material by Curran et al 2025 (<https://doi.org/10.1038/s41467-025-56154-9>)

We added it in the texts and also revised the faunal data in our dataset.

Great, it was serendipitous that this was published so you could include it! However, you

properly integrate it into your introduction, where it is mentioned in the last sentence of the first paragraph -this is out of place in the chronology you described which begins at 1.8 Ma.

Many thanks for the comment. We now have changed the text to “The first was during the Early Pleistocene, possibly ~2 Ma based on artifacts from Shangchen, China¹ and Grăunceanu in Romania dated to 1.95 Ma². From around 1.8 Ma, however, there is conclusive evidence of the presence of hominins in Eurasia^{3,4} based on fossils from Dmanisi, Georgia⁵. Shortly thereafter, stone artifacts or hominin remains are reported from the Yuanmou Formation in southern China, dated to 1.7 Mya⁶, Majuangou III and Shangshazui in the Nihewan Basin, dated to 1.6-1.7 Mya^{7,8}, Ubeidiya in the Levant dated to 1.4 Ma⁹ as well as Atapuerca and Orce in Spain, dated to 1.1-1.3 Mya¹⁰.”

Lines 47-49 – this paragraph is only a single sentence describing the 2nd OoA event, which isn't consistent with the summary of the 1st event. Could more be added for better context? *Here we concluded the driving factors of the dispersal of Homo sapiens. To keep it clear and concise, we would like to keep it this way.*

I'm afraid that it doesn't read well like this, and you need to condense and streamline the intro. Readers can't easily follow which Homo dispersal event or events you are going to focus on. You have not provided the level of detail necessary to justify discussing the second dispersal here.

We appreciate your point, but the first OoA event is subject to greater debate therefore we would like to put more detailed description on it while the second one can be briefly summarized.

Lines 64-83 – the first part of this paragraph refers to herbivores and then carnivores, so I suggest you stick with this order throughout the paragraph and review the herbivore info first, then move onto carnivores. You could then more easily link to the next paragraph (lines 84-88), which stands a bit awkwardly on its own (or better yet, incorporate it).

Here we didn't quite understand the reviewer's suggestion. The order is first all mammals, then herbivores, then carnivores, followed in the next paragraph by further discussion of carnivores.

Apologies for not being clear. You have a long paragraph covering herbivores and carnivores, followed by a two-sentence paragraph relating to carnivores that does not make sense standing on its own. You need to rewrite this part of the intro, so it is smoother.

This part was written for discussing the relationships between hominins and large mammals. The aim was to discuss herbivores then carnivores. Thanks to your suggestion, we combined and rewrote the paragraph to “Among large mammalian species, carnivores might have played a significant role in hominin dispersals¹². Hominins might have incurred in an intense competition with carnivores for both resources and space³⁰. Lewis and Werdelin (2010) proposed that large carnivores likely influenced hominin dispersal based on a significant overlapping of diet and habitat³¹. Rodríguez et al. (2023) suggested that early hominins were capable of competing with giant hyaenas for carcasses generated by saber-toothed felids³². Thus, whether hominins played a role as predators or scavengers was possibly determined by the competition stress from carnivores³⁰. Among carnivores,

Panthera gombaszoegensis, *Pachycrocuta brevirostris*^{33,34}, *Crocuta crocuta*³⁵, *Megantereon whitei*³⁴ and *Panthera leo*³⁶ dispersed into Eurasia during the Early Pleistocene^{33–36}. But whether *Pachycrocuta* and *Crocuta* originated in Africa is still unclear³⁶, and it should be taken into account that carnivores tend to have a wider geographic distribution than herbivores and can disperse quickly and widely^{37,38}

Lines 99-115 – this paragraph mostly consists of paper-by-paper one sentence summaries, rather than a synthesis. Also, and with the caveat that this is a stylistic point and perhaps the authors don't find fault with it, but nearly each sentence begins with the name of an author(s). The text shouldn't be about the people, it should be about their science. This style may be what has got you stuck in paper-by-paper summaries, as well. I would recommend that this paragraph is rewritten so that it doesn't name other researchers throughout (or that this practice is reduced) and the material is better integrated.

The paragraph has been rewritten.

It reads much better but 1) it could still synthesise the information more concisely and 2) you have added a new paragraph above it (lines 93-102 in the revised draft) that follows the same practice of sentences one after the other naming and summarising a single paper. This style is not working well for what you need to do in the introduction. At the moment, your introduction is a bit more than two pages. By contrast, the Discussion is just under two pages.

Thanks for the suggestion. The paragraph is not newly added. It was in the first manuscript we submitted. We rewrote it under the suggestion. We aimed to provide sufficient background and context in the Introduction to clearly motivate the study, while keeping the Discussion concise and focused on the key findings and their implications.

Lines 116-122 - Could this paragraph be linked more clearly with the above paragraph(s)?

This is the paragraph for introducing the hypotheses used in this study.

I understand the point of the paragraph, but my comment was about how it links to the previous text. You need to draw a connection between the intro/background and the hypotheses you want to test more explicitly.

Thanks for the suggestion, we added "based on the information above" at the beginning of the paragraph.

Line 134 – geographic distribution data comes from IUCN and although this would work for extant taxa in quite recent history, I'm not sure it works well for taxa in the deeper past, even if they are extant. For example, Thailand has had a number of local extinctions during the Pleistocene, but many of these taxa live elsewhere in Asia today (pandas and orangutan). So, the IUCN data won't provide the correct distribution data for these species. This is my first major query about the methods in this paper. I'd like to know how this issue was accounted for or if it could be (or evidence to support that there is no need to).

As noted in response to a previous reviewer's comment, in our revision we also conducted our analyses on the PHYLACINE dataset, which includes both 'current' and 'present natural' datasets. The latter estimates geographic ranges in the absence of human impacts. Results of the PHYLACINE analyses are similar to those using IUCN data (shown as

Supplementary Fig. 4)

This is a great way to address the issue I highlighted. However, you have not explained why you use the PHYLACINE dataset in the paper. Explain the potential problem with using IUCN data, and what the PHYLACINE data is based on that helps you get around the IUCN problem.

We added explanation for using the PHYLACINE dataset in “Methods” section: “The Phylacine dataset includes species ranges and diversity estimated while accounting for large human impact”.

Lines 145-147 – are these the same diet and locomotion categories used by others, so you are following an established system?

Yes. The sources of the trait scores are listed in the same paragraph. We also added the reference in the supplementary spreadsheet file “trait.csv

My apologies, I didn’t quite describe my question very well! You do note the sources of your diet and locomotion/spatial categories and say that your trait data are “based on” them (lines 149-151). Are they FROM these sources or BASED ON them? This is an important distinction. If they are based on them, how did you make decisions about which categorization systems to use, or how to reconcile them from different sources? What did you do to ensure that you were consistent in your categorisations when the data came from multiple sources?

We changed “based on” to “from”. We did not decide the categories but only extracted the info from different datasets or literatures. We added “If a genus has species in different diet categories, all the categories will be assigned to the genus.” to explain how to deal with this situation.

Line 149-150 – It would be more succinct to simply state what the analysed body categories were rather than explain that small mammals were excluded after their body size categories are given earlier in the paragraph.

For genera that span more than one size category, the exact value is determined by randomly choosing a single value early in the analysis. The exclusion of the body mass categories 1 and 2 therefore needed to be done after the randomized choice of the body mass (all shown in the R script provided).

Ok, that makes sense, but it needs to be explained in the main text, or you need to find a way to avoid a reader asking the question, as I did. You can’t assume that readers will look over the R code to learn the answer.

We now changed the order in the text and explain that the exclusion of the body mass is done after the randomization choice.

Lines 158-160 – how did you find that $k=3$ is the minimum number of clusters needed to distinguish the continents? That rather makes intuitive sense since there are three continents, but the evidence to support this statement should be provided. Is it in the supplementary info, but I missed it? This is my second major methods-related question. Also, the supplementary info should be referred to wherever it includes data not summarized/presented in the main text. There was a lot of missing info such as the names

of the sites, their constituent fauna, their geological dates, their references, and info about the trait assignments for each genus, and those references. There are files pertinent to these data available to download, but they should be referred to in the text.

We now conduct a silhouette coefficient analysis for determining the value of k . The best value of k for fossil data is 2 and 3 respectively for taxonomic and functional clusters. The best value for extant data is higher. But our aim was to compare the differences between the extant and fossil data, so we determined that all values of k should be 3. We provided more plots in the Supplementary as s supplementary material. Additionally, we added the reference for the supplementary material in the main text.

This is an interesting approach, but I am still not following the logic of selecting $k=3$. Please bear with me! The silhouette coefficient analysis results say that the best value of k is 2 for fossil functional clusters, and the AMD results in 2 for fossil taxonomic clusters.

The extant clusters do not seem to yield higher values of k as you say above; it is 3 in all cases, except for extant functional clusters which is 2. I am looking at Supplementary Fig 1 and this is what the caption describes. Three out of eight analyses say that $k=2$ and five out of eight say that $k=3$, so how do you then land on 3 being “best”? Neither the main text nor the supplementary figure explain this clearly.

In supplementary Fig1, analyses shown in panels A, C, F, G, and H actually showed a best number of 3. So, it is 5 out of 8 showing that the best number should be 3. Based on this and based on our hypotheses which considered the possibility that three continents might have had three separate faunas, we use $k = 3$ for all the analysis in our study.

Lines 210-216 – I must confess that I am not that familiar with chronofauna analysis and that I struggled to get my head around the purpose of it here. I have to ask if it is necessary to the overall story. In looking at the results in figure 2 it looks like the Asian and European fauna follow the exact same curve and that they remain the same distance apart until just before 2.5 million years ago when they diverge from each other at the start of the Pleistocene. So, they both grow increasingly dissimilar to SSMZ until their curves diverge and Asia starts to look more similar to it. This isn't commented on, but it merits an explanation, particularly the timing. This is my fourth major comment on the methods.

Good point. The chronofauna visualization is in reference to a single site, and for our broader continental-scale perspective it is not necessarily relevant to describe the differences of the geographic curves in detail. Furthermore, we have now added three further site comparisons. The goal of these visualizations is to provide further support to the distinctiveness of the African assemblages in relation to the Eurasian ones, and to show the closer similarity and turnover of European and Asian assemblages among each other.

Adding other reference sites helps make the point of the chronofauna analysis clearer, and addresses another reviewer's suggestion that the analysis be extended. In looking at the new results now, though, I wonder if they are actually complicating the point you want to make - certainly you need to be more explicit in describing what these results show in relation to your paper's goals. Many studies show that that African fauna is different from Eurasian fauna in the past and present, so what does this demonstration do to support your hypothesis testing (which is also specifically related to a much more specific time

period)?

Actually, studies describing the distinctiveness of African and Eurasian fossil faunas are very few, and to our knowledge there is no study of the scope we have undertaken. In exploring the larger context of African-Eurasian biogeographic similarity, chronofaunal analysis provides an added perspective. It reveals further details of geographic similarity from the perspective of the specific reference site chosen. In this case, it confirms the broader historical patterns of the taxonomic clustering from four geographically distinct, site-specific perspectives.

Lines 230-238 – it wasn't until these results of the correspondence analyses that I realised I was unclear about how the time variable was considered in them. I don't think it was accounted for, am I right in this? If that is so, everything has been lumped together over a 10-million-year period...? To me, that doesn't make sense. Firstly, it is known that taxonomically these three continents differ from each other, but that Europe and Asia share more similarities. So, this doesn't seem entirely new, but a confirmation of what is known - but it would be insightful if the differences were explored over time to look for more nuanced trends (although I suspect there would be a high level of differentiation throughout the time sequence). Secondly, my feeling is that lumping the trait data together for 10 million years was likely to result in a lack of differentiation between the three continents, and again there would be more nuance and insights if the data were analysed by time bin. This is my fifth major methods comment.

Yes, the fossil data are analyzed together in the correspondence analysis. The reason for this is to provide visual information on which taxa and traits have the strongest influence on (i.e. most strongly define) the three clusters determined by the clustering analysis. Analyses by 1 myr time bin, whether for taxonomic or functional data, are not feasible at this scale of analysis as the data would be far too few. An analysis at this smaller timescale would also require using species-level taxonomy and a more refined set of traits might be able to do this (e.g. Faith et al. 2019 www.pnas.org/cgi/doi/10.1073/pnas.1909284116)

Ok, that explanation makes sense, but it is not present in the manuscript. The text relating to correspondence analysis in the methods section is overly general. You need to explain what you have described above – what is the purpose of this analysis in light of your study's goals?

We added the explanation into the method section: "Additionally, correspondence analyses of the community-genus and community-trait data matrices were conducted to visualize the relationships between communities, genera, and traits for providing visual information on which taxa and traits have the strongest influence on (i.e. most strongly define) the three clusters determined by the clustering analysis. Correspondence analysis is a multivariate statistical technique used for visualization of the relationships between multiple categorical variables."

Line 298 – "Data inaccuracy is unlikely..." That is a strong statement! There are surely a host of inaccuracies in the data which have not been acknowledged (which are not the authors' fault, but simply the reality of working with fossil fauna community data)...

taphonomy, depositional context, difficulties with genus identification, sampling etc. Aspects of the methods could also confound the analyses – for example, the dietary categories are blunt, and include the category ‘omnivore’ which is something of a catch-all, and sites with only five unique genera were included, which is a very small number. Other researchers have grappled with this latter issue, Louys et al. 2009 finding that 12 taxa was an acceptable minimum number (<https://doi.org/10.1016/j.jas.2009.06.012>).

Our point was that assigning body mass, diet, and locomotion using our coarse categories was largely uncontroversial for the majority of the taxa. We reworded this part to try make this clearer: There are possibly two explanations for these differences: 1) The fossil datasets may be severely inaccurate or incomplete; 2) A fundamental rearrangement of functional groups appeared in the time between the fossil and modern dataset, i.e. during the Late Pleistocene and Holocene. We think it is unlikely that this pattern is the result of inaccurate data, particularly given that the attribution of such fundamental traits as size, diet, and locomotion to both fossil and extant genera was in mostly cases done with a relatively high degree of confidence. Also in Louys et al. 2009, their method and aim were quite different from our study. They determined a minimum number of species for distinguishing different habitats - which is not an objective of ours. As a comparison, Kaya et al. 2018 (<https://doi.org/10.1038/s41559-017-0414-1>) also used a minimum of five taxa identified to the genus level for their biogeographic similarity analyses

I would strongly disagree that the use of the term “omnivore” is uncontroversial. You do not define it in your paper, but point readers only to the sources of your trait data, which does not explain how your study has used the info in the sources (I noted this earlier – did they provide different info on some taxa, or use different trait classification systems to your own?). Omnivore is a catch-all term, and a poor one at that because it does not have a stable definition. Another reviewer gave a great example of this with respect to bears. By “omnivore”, do you simply mean any animal that will eat a combination of vegetation and animal matter? What combination of these resources is the threshold for inclusion in the “omnivore” category? Given how broad the categories “herbivore” and “carnivore” are, many mammals could be classified as omnivorous (many herbivores eat invertebrates or eggs, for example and many carnivores do not hesitate to snack on wild fruit). Similarly, the locomotion categories mix both actual locomotion behaviours and the space occupied by a species, and these categories come with their own controversies (in your case, the use of the term amphibious was a bit curious – it’s synonymous with the term semi-aquatic which is more commonly used, in my experience...?). These are all well-known conundrums, but they just be addressed. The use of the term omnivore may have an enormous impact on your clustering – so how is it justified? I personally try to avoid its use as I don’t believe it is a meaningful classification, but if you want to use it, it needs to be clearly defined and justified. Saying that there was high confidence in the attribution of fossil taxa to categories, or that the process of categorisation is straightforward, comes across as naïve.

We understand that omnivore is a poorly-defined term. As mentioned above, dietary categories were taken from other studies that made these determinations. Perhaps more

importantly, redefining omnivore and revisiting these categorizations is unlikely to affect the results of our analyses. In our study, there are only 46 out of 625 genera (7.4%) classified as omnivores, while 32 out of 625 have omnivore in their diet category. Nonetheless we do now mention the problematic nature of the omnivore category, and also its limited proportion in our data in our Method section.

With respect to the number of genera at each site included in your analyses, I understand your point and see that using five is a cut-off was sensible. However, you do not cite the Kaya reference in the methods section or explain the rationale, both of which should be added.

We added the Kaya et al. reference to the text

NEW COMMENTS ON REVISED DRAFT:

Figure 1 – the legend still needs to be more explicit with respect to what clusters 1, 2 and 3 represent. Give them better/descriptive names in the legend. The caption has added some more detail in line with another reviewer's request, but all of the clusters were not described, only some of them.

We added more text in the caption to explain the clusters. For the legend, we prefer to keep the clusters named by number, for simplicity.

Line 56 – use a more scientific word than “thrived”

We have changed it to “evolved”

Line 133 – readers should be referred to the table where the published literature/references can be found.

It is referred to the table 1(fossil.csv) in line 136 after introducing the whole table.

Line 139 – grid cells are not the same thing as communities, so either use the term grid cell or explain that the word community used here refers to each cell

We added “Hereafter, *grid cells* are referred to as *community* throughout this paper.” to explain in the text.

Line 146 – groups, not levels

changed.

Line 155-157 – where can readers find these analyses and results?

The randomization of the data can be done in the code provided. It makes would not make much sense to provide random results to the readers. We added that the code is available to use and run the data in our text instead.

Line 165 – delete “in this study”

Deleted

Line 166 – a list of three matrices?

We changed to text to “The result consisted of three dissimilarity matrices, and total dissimilarity was selected in this study.”

Line 173 – refer readers to the relevant figure/supplementary section

Added

Line 241-242 – older and younger, not old and new

Changed

Line 243 – dominated not -ing

Changed.

Line 290 – rephrase “started to become low again”

We rephrased the whole paragraph and now it is “The chronofauna analyses of other communities from the Pleistocene of Spain (Atapuerca TDW4 & TDE5), Miocene of Anatolia (Akkasdagi), and Pleistocene of Kenya (Karari Ridge 2) confirm the high similarity of Asian and Eurasian faunas during the Late Miocene (Akkasdagi) and their later divergence during the Pleistocene (Atapuerca), and the highly endemic nature of African Plio-Pleistocene faunas (Karari Ridge) (Figs 1B-D).”

Line 302-303 Figure 3 caption – this sentence isn’t clear.

The sentence in the caption was changed to “Analyses were conducted with all the data then visualized into the three different epochs”.

Also, explain in the text why the analysis of every palaeocommunity from a 10 million year period is useful for you and include a statement about this in the caption as well. The results themselves aren’t surprising or new, so readers need to be told why you have done this.

The reason for the study and the dataset is discussed at length in the Introduction. We do not think it would make much sense to explain it explicitly here in the caption of the nMDS analysis. Furthermore, we do not agree that the results shown in this figure (which support those in other figures, which use different techniques), aren’t surprising or new. Neither the deep and long-term taxonomic separation of Africa from Eurasia, nor the lack of functional community structure in fossil data were expected or demonstrated before.

Line 320-321 – medium sized mammals are the most common large mammals? That doesn’t make sense. More to the point, an animal weighing 10,935 kg is definitely not medium sized. How are distinguishing between mediums and large mammals?

You are correct. It is confusing here. We changed it to “Ground dwelling herbivores between 45 kg and 10 tons are the most common large mammals in all three continents.”

Figure 4 – why is every 20th entry selected for visualisation? I don't understand why this was done - it is not a standard way to describe a plot of CA results. This should either be justified or modified.

There are too many genera in our study to plot in a single figure (it would be unreadable). Reducing these to 1 in every 20 labels keeps the figure readable. We have simplified the caption now to say: Only a subset of genera is labeled for readability

Line 345 – tiny is not an appropriate word here.

We changed to minor.

Line 380 – this section needs to acknowledge that no changes to community structure were detected with the very coarse trait categories studied. The scale is relevant.

We mentioned that it is under the trait categories used in our data in this section. The scale is also mentioned in the Method section. We mentioned the trait category in the latter section “Evidence for highly altered extant functional community structure” where we compared the results of the fossil and extant functional analysis.

I hope my comments on the revised draft are useful to you and, as always, I am happy for you to ask for help if any of my points are unclear!

Kris Kovarovic
Durham, UK

Many thanks for all the suggestions.